# Cathodal weak direct current decreases epileptic excitability with reduced neuronal activity and enhanced delta oscillations

Chia-Chu Chiang[1,2], Miao-Er Chien[3], Yu-Chieh Huang[3], Jyun-Ting Lin[4], Sheng-Fu Liang[4], Kuei-Sen Hsu[5,6] (iD), Dominique M. Durand[1,2] (iD) and Yi-Jen Wu[3,7] (iD)

[1]*Department of Biomedical Engineering, Case Western Reserve University, Cleveland, Ohio, USA*

[2]*Department of Biomedical Engineering, Neural Engineering Center, Case Western Reserve University, Cleveland, Ohio, USA*

[3]*Institute of Clinical Medicine, College of Medicine, National Cheng Kung University, Tainan, Taiwan*

[4]*Department of Computer Science and Information Engineering, National Cheng Kung University, Tainan, Taiwan*

[5]*Department of Pharmacology, College of Medicine, National Cheng Kung University, Tainan, Taiwan*

[6]*Institute of Basic Medical Sciences, College of Medicine, National Cheng Kung University, Tainan, Taiwan*

[7]*Department of Neurology, National Cheng Kung University Hospital, College of Medicine, National Cheng Kung University, Tainan, Taiwan*

Handling Editors: Katalin Toth & Gareth Morris

The peer review history is available in the Supporting Information section of this article (https://doi.org/10.1113/JP287969#support-information-section).

*The Journal of Physiology*

**Abstract figure legend** Polyspike epileptiform discharges and excitatory unit spike activities are observed in rats with kainic acid (KA)-induced hippocampal seizures. A weak electric field generated by cathodal direct current stimulation (ctDCS) at CA1 reduces seizure excitability by decreasing the frequency and amplitude of epileptic spikes and increasing $\delta$ power in local field potentials. After ctDCS, the excitatory neuronal firing is diminished, accompanied by a strengthened coupling between unit spikes and $\delta$ waves. These effects last up to 90 min post-stimulation and are associated with short-term synaptic plasticity changes, as demonstrated by paired-pulse stimulation. Hippocampal brain-derived neurotrophic factor (BDNF) and neuronal activation, measured by *c*-Fos, significantly decline after ctDCS in CaMKII[+]-excitatory neurons while increase in GAD[+]-inhibitory neurons. This study elucidates how ctDCS alleviates epileptic excitability by modulating neuronal activity and enhancing endogenous $\delta$ oscillations through strengthened unit spike–$\delta$ wave coupling.

C.-C. Chiang and M.-E. Chien have contributed equally to this work.

**Abstract** Seizures are manifestations of hyperexcitability in the brain. Non-invasive weak current stimulation, delivered through cathodal transcranial direct current stimulation (ctDCS), has emerged to treat refractory epilepsy and seizures, although the cellular-to-populational electro-physiological mechanisms remain unclear. Using the ctDCS *in vivo* model, we investigate how neural excitability is modulated through weak direct currents by analysing the local field potential (LFP) and extracellular unit spike recordings before, during and after ctDCS versus sham stimulation. In rats with kainic acid (KA)-induced acute hippocampal seizures, ctDCS reduced seizure excitability by decreasing the number and amplitude of epileptic spikes in LFP and enhancing delta ($\delta$) power. We identified unit spikes of putative excitatory neurons in CA1 stratum pyramidale based on waveform sorting and validated via optogenetic inhibitions which increased aberrantly in seizure animals. Notably, cathodal stimulation significantly reduced these unit spikes, whereas anodal stimulation exhibited the opposite effect, showing polarity-specific and current strength-dependent responses. The reduced unit spikes after ctDCS coupled to $\delta$ oscillations with an increased coupling strength. These effects occurred during stimulation and lasted 90 min post-stimulation, accompanied by inhibitory short-term synaptic plasticity changes shown in paired-pulse stimulation after ctDCS. Consistently, neuronal activations measured by $c$-Fos significantly decreased after ctDCS, particularly in CaMKII$^+$-excitatory neurons while increased in GAD$^+$-inhibitory neurons. In conclusion, epileptic excitability was alleviated with cathodal weak direct current stimulation by diminishing excitatory neuronal activity and enhancing endogenous $\delta$ oscillations through strengthened coupling between unit spikes and $\delta$ waves, along with inhibitory plasticity changes, highlighting the potential implications to treat brain disorders characterized by hyperexcitability.

(Received 28 October 2024; accepted after revision 12 March 2025; first published online 7 April 2025)

**Corresponding authors** D. M. Durand: Department of Biomedical Engineering, Case Western Reserve University, Cleveland, OH 44106, USA. Email: dxd6@case.edu

Y.-J. Wu: Institute of Clinical Medicine, College of Medicine, National Cheng Kung University, Tainan, Taiwan, and Department of Neurology, National Cheng Kung University Hospital, Tainan City 70457, Taiwan. Email: wuyj@mail.ncku.edu.tw

**Key points**

- Electric fields generated by transcranial weak electric current stimulation were measured at CA1, showing polarity-specific and current strength-dependent modulation of unit spike activity.
- Polyspike epileptiform discharges were observed in rats with kainic acid (KA)-induced hippocampal seizures. Cathodal transcranial direct current stimulation (ctDCS) reduced the number and amplitude of the epileptic spikes in local field potentials (LFPs) while increased $\delta$ oscillations.
- Neuronal unit spikes aberrantly increased in seizures and coupled with epileptiform discharges. ctDCS reduced excitatory neuronal firings at CA1 and strengthened the coupling between unit spikes and $\delta$ waves.
- Neuronal activations, measured by $c$-Fos, decreased in CaMKII$^+$-excitatory neurons while increased in GAD$^+$-inhibitory neurons after ctDCS.
- These effects on LFP and unit spikes lasted up to 90 min post-stimulation. Inhibitory short-term plasticity changes detected through paired-pulse stimulation underpin the enduring effects of ctDCS on seizures.

# Introduction

Non-invasive application of weak direct currents through transcranial direct current stimulation (tDCS) is widely used to modulate brain functions (Weller et al., 2020; Wu et al., 2014) and cognitions in patients with neuro-psychiatric disorders (Ko et al., 2022; Lu et al., 2019; Wu et al., 2016). A subthreshold electric field (EF) at the human cortex induced by a weak current of 1 mA tDCS is estimated as 1 mV/mm, which can modulate cortical excitability with polarity-specific stimulation effects (Jackson et al., 2016). In general, anodal tDCS

generates a facilitatory effect, whereas cathodal tDCS (ctDCS) causes an inhibitory effect, although non-linear effects occur under certain conditions (Nitsche et al., 2005). tDCS modulates neuronal polarization along the parallel orientation to the induced EF and changes the synaptic plasticity (Rahman et al., 2013; Wu et al., 2017). In hippocampal slices, ctDCS has been reported to hyperpolarize the neuronal membrane potential to modulate excitability (Chang et al., 2015; Voroslakos et al., 2018). Seizures arise from aberrant excitatory neuronal activities and may evolve to persistent repeated seizures refractory to medication control when the brain loses its self-termination mechanisms (Betjemann & Lowenstein, 2015). Taking advantage of the inhibitory effects and non-invasiveness, ctDCS holds promise in reducing seizures in patients with refractory epilepsy (Fisher et al., 2023; San-Juan et al., 2017; Simula et al., 2022; Yang et al., 2020). Clinical heterogenicities of subjects and protocols have resulted in varying outcomes of tDCS treatment and necessitate mechanistic studies (Sudbrack-Oliveira et al., 2021). tDCS delivers weak direct currents, which is the subthreshold for generating action potentials, and it differs from alternating current entraining the neurons through the applied frequency (Huang et al., 2021). There remains a gap in understanding the *in vivo* electrophysiological mechanisms by which the weak direct current reduces seizure excitability through neuronal-to-populational modulations.

Our animal study has shown that repeated ctDCS decreases severe convulsive seizures and reduces gamma ($\gamma$) power on local field potentials (LFPs) in a rat model of status epilepticus (Wu et al., 2020). In rats with chronic spontaneous seizures, delta ($\delta$) oscillations were positively correlated with a reduction in interictal spikes, suggesting the involvement of low-frequency oscillations in the inhibitory effects of tDCS on seizures (Wu et al., 2021). Congruently, intracranial electroencephalogram recordings of the patients with epilepsy revealed shorter durations of the ictal-epileptic discharges during tDCS treatment with increased power density over $\delta$, $\theta$ and $\alpha$ bands compared to those without tDCS (Koessler et al., 2023). However, the specific influence of the weak direct current on populational oscillations through modulating neuronal excitability in seizure is elusive. Here, we investigate the neuronal bases underlying the LFP changes after ctDCS in an animal model of hippocampal seizures. Seizures are initiated from neuronal hyperexcitability and propagated with devastated inhibition. Excessive excitation of pyramidal cells, resulting from the breakdown of feed-forward or feed-back inhibition within microcircuits, can generate seizures (Paz & Huguenard, 2015). Certain interneurons have been implicated to play an excitatory role in epileptogenesis influencing the epileptic circuitry by synchronizing principal neurons or disinhibiting pyramidal cells through the suppression of other inhibitory interneurons (Toyoda et al., 2015; Ye & Kaszuba, 2017). Hippocampal rhythms are modulated by interneurons and pyramidal neurons underlying various cognitive functions (He et al., 2021). There is a potential link between neuronal firing activity and hippocampal rhythm where the neuronal unit spikes couple with LFP and alter the regional oscillations in seizures. The study aims to uncover the neuronal-to-populational electrophysiological mechanisms through which epileptic excitability is modulated by weak direct current stimulation. We hypothesized that ctDCS reduces excitatory neuronal activity, thereby entraining endogenous LFPs into low-frequency oscillations through the coupling between unit spikes and LFP.

## Materials and methods

### Animal surgery, kainic acid-induced seizures and tDCS protocol

Male Sprague–Dawley rats (BioLASCO, TPE, TW), 6 weeks old, were placed in a stereotaxic instrument (RWD, CA, USA) for surgery under anaesthesia with 2%–2.5% isoflurane inhalation and with a pre-surgical analgesia of ketoprofen (2 mg/kg, intraperitoneal). A plastic cannula (1 cm height, 1.59 mm inner diameter) was fixed to the skull at the AP coordinate −3 mm, which was filled with 0.9% normal saline to serve as a socket for the cathodal stimulation electrode (0.6 mm-diameter gold-plated copper alloy pin electrode, Nan E. Electronic Co., TNN, TW). An anodal electrode

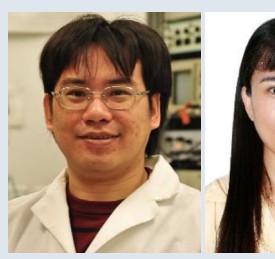
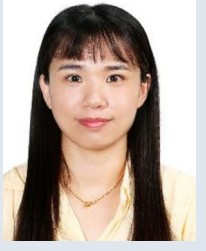

**Chia-Chu Chiang** is a research assistant professor in the Neural Engineering Centre (NEC) and Department of Biomedical Engineering at Case Western Reserve University. His research interests are in epilepsy control by neural modulation and studying the mechanisms of seizures in neural networks. **Miao-Er Chien** is a research technician in Dr Yi-Jen Wu's Lab at the National Cheng Kung University. Working with Dr Wu, she focuses on exploring the therapeutic potential and neurophysiological mechanisms of electrical neuromodulation for seizures and epilepsy.

(10 mm-diameter gold-plated cup electrode, Technomed Europe, NL) was applied at the dorsal shoulder when stimulated with a conductive gel (Fig. 2*A*; Wu et al., 2021). CA1 receives excitatory inputs from the entorhinal cortex through the perforant pathway and from CA3 via the temporoammonic pathway, and is susceptible to seizure induction within the hippocampus (Mueller et al., 2023). A tungsten recording microelectrode (127 μm diameter, A-M System, WA, USA) was placed in dorsal hippocampal CA1 at AP: −5, ML: 3.5 and DV: −3 mm for signal recording. Two stainless-steel screws (M1, 4 mm length) were placed at ML: 0, AP: −6 for reference and AP: +1.5 mm for ground. One 31 gauge microinjection needle was inserted into CA3 at AP: −5.67, ML: 1.57, DV: −6.5 mm, 30° with vertical axis and attached to a laboratorial syringe pump (Langer Instruments, AZ, USA) for kainic acid (KA) infusion. KA (Abcam, Cambs, US), a selective agonist for kainate glutamate receptors, was infused at a dose of 0.4 μg in 0.2 μl normal saline, with an infusion rate of 0.1 μl/min to induce hippocampal seizures (Levesque & Avoli, 2013). ctDCS was applied 2.5 h after KA infusion when the epileptiform discharges evolved to sustained polyspikes. Rats were subjected to cathodal constant direct current through the epicranial stimulation electrode above the dorsal hippocampus with 1 mA (ctDCS) or 0 mA (sham) for 30 min stimulation with a Master-9 stimulator and an ISO-Flex stimulus isolator (A.M.P.I., HSZ, TW). The experimental animals were housed with *ad libitum* access to food and water. All experimental procedures were in accordance with the National Institutes of Health Guidelines for the Care and Use of Laboratory Animals and approved by the National Cheng Kung University (NCKU) Institutional Animal Care and Use Committee (approval no. 109016).

### Signal acquisition and EF measure

Signals were collected through a 0.7–8 kHz bandpass filter with a 20 kHz sampling rate and a gain of 800 using a tethered record system (Triangle BioSystems International) and an MP150 acquisition system (Biopac Systems Inc., CA, USA). Signals for unit spikes were filtered using a high-pass filter of 300–3000 Hz, whereas signals for LFPs were processed with a down-sampling rate to 2 kHz. To maintain the stability in unit spike recording during acute seizures, signals were collected under anaesthesia with 1%–1.5% isoflurane inhalation. Signals were analysed offline using MATLAB software (MathWorks, MA, USA). For EF measure, a twisted pair of double-wire electrodes (203 μm-diameter stainless-steel wire, A-M System, WA, USA) with a 1 mm distance between the tips of two recording electrodes was inserted at the dorsal hippocampus CA1 (AP: −3, ML: 1.5, DV: −3 and −2 mm). Low-frequency sinusoidal stimulations

were applied using a Stmisola linear isolated stimulator (Biopac Systems Inc., CA, USA) through the epicranial stimulation electrode to test the varying transcranial current strength-induced EF at CA1. The field potentials (mV) were calculated as the difference between the paired electrode tips divided by the distance of 1 mm, resulting in the EF (mV/mm) measured at CA1 (Fig. 2*B*).

### Epileptic spike detection, calculation and power analysis in LFPs

LFP primarily reflects the synaptic potentials pooled across populations of neurons. The 30-min LFP signals before seizure induction were used as the baseline. Peaks with amplitudes greater than 10 times the baseline average and an absolute amplitude ≥1 mV and width ≤140 ms were labelled as epileptic spikes in LFPs using the epileptic spike detection algorithm (Appendix Methods). The signals were clear without stimulation artifacts from the weak direct current stimulation, for which no additional template subtraction for artifacts was implemented. After spike detection, the number and amplitude of the identified epileptic spikes were analysed. The amplitudes of the spikes were calculated by integrating the spike count and amplitude information using the areas under the curve (AUCs) of the spike amplitude distribution, which represents the spike amplitudes and their corresponding spike counts. The AUC was then normalized to the pre-stimulation stage for each animal individually to compare amplitude changes across the pre-stimulation, stimulation and post-stimulation stages in the ctDCS and sham groups, respectively. The frequency-band power of each period was calculated using a spectrogram with a short-time Fourier transform. The LFP signals were processed using MATLAB's built-in Welch's method for power spectral density (PSD) using 2000-point segments with 50% overlap. LFP epileptic spikes and power comparisons between ctDCS and sham groups were measured from the pre-stimulation (pre-stim), stimulation (stim) and post-stimulation periods of 0–30 min (P0–30), 30–60 min (P30–60) and 60–90 min (P60–90), respectively, each containing 30-min signal segments. Signals for PSD were sampled from 3–5, 8–10, 13–15, 18–20, 23–25 and 28–30 min over each period of pre-stim, stim and post-stimulation (post-stim) in ctDCS- and sham-treated animals.

### Unit spike sorting

*In vivo* extracellular unit spike recording is used to measure the changes in electrical current outside a neuron when an action potential is generated by individual neurons. Signals for unit spike sorting were sampled from 3–5, 8–10, 13–15, 18–20, 23–25 and 28–30 min

over each period of pre-stim, stim and post-stim in ctDCS- and sham-treated animals. The unit spike signals were recorded at CA1 stratum pyramidale from animals under anaesthesia and sorted using the UltraMegaSort 2000 algorithm (https://neurophysics.ucsd.edu/software.php) based on waveform similarity using k-means clustering and firing features analysis, incorporating autocorrelation and cross-correlation (Hill et al., 2011). The output was then imported into MATLAB, where clusters were manually edited using custom spike sorting algorithms based on waveform similarity and spike width. Unit spikes of the clearly separated clusters were analysed. The spike width of the hippocampal pyramidal neurons and interneurons in rodents was reported, with peak-to-peak spike widths ranging from 0.59 to 0.86 ms for pyramidal neurons and 0.35 to 0.45 ms for interneurons (Csicsvari et al., 1998; English et al., 2017). Unit spikes with a peak-to-peak spike width of 0.6–0.7 ms were observed in patients with epilepsy during seizures (Merricks et al., 2015, 2021). Unit spikes with a width of 0.5–1 ms were identified as putative excitatory neurons in this study. The *in vivo* unit spike numbers were counted for frequency. The changes in spike number were calculated using the formula for both unit spikes and LFP spikes:

$$\frac{\Delta N}{N} = \frac{N(t) - N(0)}{N(0)}$$

where $N(t)$ is the spike number at a given time period and $N(0)$ is the mean value of the spike number at the corresponding pre-stimulation time period in individual rats.

## Coupling strength calculation

To calculate the coupling between unit spikes and $\delta$ oscillations, the phase values of $\delta$ wave ($\geq 0.5$ to $< 4$ Hz) were extracted and converted to the phase degree using Hilbert transform. The distributions of unit spikes were allocated to the underlying $\delta$ waves based on the phase degree. The number of unit spikes at each $\delta$ phase was divided by the total number to calculate the proportion of unit spikes in each $\delta$ phase. The unit spike–$\delta$ wave coupling strength was calculated using the equation for the phase-lock value:

$$\text{Coupling strength} = \left| \frac{1}{n} \sum_{k=1}^{n} e^{i\emptyset_k} \right|$$

where $n$ is the number of unit spikes in each spike train, $\emptyset_k$ denotes the instantaneous phase of the $\delta$ wave determined by Hilbert transform at the same time as the $k$th spike and $e^{i\emptyset_k}$ determines the complex exponential function of $\emptyset_k$ following Euler's formula (Davis et al., 2020).

## Optogenetic inhibition *in vivo*

pAAV-Ef1a-DIO eNpHR-EYFP viruses (Addgene, 26966, MA, USA) were transfected into Vgat-ires-cre mice (Slc32a1[tm2(cre)Lowl/]J mice, The Jackson Laboratory, 016962, ME, USA), whereas pAAV-CaMKIIa-eNpHR3.0-EYFP viruses (Addgene, 26971) were unilaterally injected into C57BL/6J male mice, 6–10 weeks old (NCKU animal centre), at CA1 (AP: −3, ML: 2.5, DV: −1.6 mm) under anaesthesia with 2%–2.5% isoflurane inhalation and pre-surgical analgesia with ketoprofen (3 mg/kg, intraperitoneal). After 10 days of recovery a 50.8 μm-diameter tungsten recording wire (California Fine Wire Co., CA, USA) attached to the optic fibre (high OH for 300–1200 nm, TECS Clad, Thorlabs, NJ, USA) was implanted into CA1 (AP: −3, ML: 2.5, DV: −1.6 mm) for optical stimulation using a LED driver (PlexBright LD-1, Plexon Inc., TX, USA). KA was then microinjected into CA3. After 2.5 h of KA induction, CA1 was subjected to a 590 nm laser light with an 8 mW intensity of 10 s-on and 10 s-off pulses, repeating 30 cycles for 10 min. The unit recording signals were recorded under anaesthesia with 1%–1.5% isoflurane inhalation at pre-optostim (5 min), optostim (10 min) and post-optostim (5min) periods through a high-pass filter of 300–3000 Hz at a sampling rate of 20 kHz using a digitizing headstage and the acquisition system for offline analyses.

## Paired-pulse stimulation *in vivo*

Paired-pulse stimulation (PPS) was applied using a stainless-steel wire stimulation electrode (254 μm diameter, A-M System) with a −1.5 mA pulse stimulation in a 0.1 ms pulse-width square wave to the left paramedian side of the ventral hippocampal commissure (AP: −0.7, ML: −0.7, DV: −4.5 mm), alongside a stainless-steel screw return electrode (AP: +0.3, ML: −0.7 mm). Evoked responses were recorded at the left CA1 (AP: −2.5, ML: −2.5, DV: −3 mm) under anaesthesia with 1%–1.5% isoflurane inhalation using a tungsten recording microelectrode (127 μm diameter, A-M System). The *in vivo* PPS protocol was applied with interstimulus intervals (ISIs) of 3.33, 5, 10, 20, 40 and 100 ms (Jensen & Durand, 2009). PPS was repeated 10 times for each ISI with a 0.3-s interval between each pair. The amplitudes of the first (P1) and second (P2) evoked potentials were calculated for the P2/P1 ratio to evaluate the hippocampal excitability change and short-term plasticity. When the ISI was <20 ms, P1 was obscured by the second stimulation pulse. In these cases, the mean responses from 10 P1 traces at an ISI of 100 ms in the same rat were used as a substitute for P1 (Fig. 5*A*).

## Immunofluorescence and brain-derived neurotrophic factor assays

Immunofluorescence was performed as in the previous report (Wu et al., 2020). After recording, the rats were sacrificed to obtain brain slices at 4.5 h after KA injection. Hippocampal coronal sections (40 μm thick) were incubated with primary antibodies against c-Fos (1:500, Cell Signaling Technology, mAb2250, MA, USA), calcium/calmodulin-dependent protein kinase IIa (CaMKIIa, 1:200, Abcam, Ab22609, Cambs, UK), glutamic acid decarboxylase 67 (GAD 67, 1:50, Sigma-Aldrich, G5419, MO, USA) and NeuN (1:2000, Millipore, ABN78, UK) overnight at 4°C in blocking solution (Dako, S3022, CA, USA). The sections were then washed in PBS with 0.4% Triton and incubated with Alexa Fluor 488- and Alexa Fluor 568-conjugated secondary antibodies (A-11008, A-11004, Thermo Fisher Scientific, MA, USA) for 2 h at room temperature. Nuclei were visualized using a mounting medium with DAPI (Abcam, Ab104139, Cambs, US). Fluorescence microscopic images for neurons were obtained using an Olympus FluoView FV3000 confocal laser scanning microscope and analysed using TissueQuest 4.0 by the operator blinded to experimental conditions. The neuronal activations examined using immunofluorescent staining were sampled from the dorsal hippocampal CA1 and dentate gyrus (DG) below the stimulation cannula (stim site, AP: −3 mm) and the electrode recording sites (record site, AP: −5 mm). Hippocampal brain-derived neurotrophic factor (BDNF) collected from the dorsal hippocampal area under the stimulation electrode was analysed for the concentrations using a conventional ChemiKine BDNF Sandwich ELISA kit (Millipore, UK). After the experiments were completed, all animals were euthanized using 70% $CO_2$ administered for at least 5 min until the heartbeat and breathing ceased.

## Statistical analysis

Results are reported as median $\pm$ interquartile range (IQR) for non-normally distributed data and as mean $\pm$ standard deviation (SD) for data that are approximately normally distributed. Statistical analyses were conducted using Prism software. Unpaired $t$ tests were applied to compare normally distributed data of the two groups. Non-parametric statistics using Mann–Whitney $U$ test and Kruskal–Wallis test were used to compare non-normally distributed data. Repeated-measures two-way ANOVA assessed the main factors and the interactions, followed by *post hoc* analysis using Bonferroni multiple comparisons test. Significance was set at $p < 0.05$.

## Results

### Neuronal unit spikes coupled with epileptiform discharges in LFPs

To determine the relationship between neuronal firing and epileptiform discharges, we analysed extracellular unit spikes and LFPs simultaneously recorded from CA1 in awake animals with CA3 KA-induced seizures. Epileptic activities appear in distinct patterns during seizure evolution after KA induction, prior to reaching status epilepticus. Once status epilepticus is achieved, the LFPs predominantly exhibit sustained high-frequency polyspikes. We identified three epileptiform discharge patterns in LFPs as polyspike, spike-and-wave and mixed patterns. Polyspikes consisted of high-amplitude signals in the $\gamma$ range (Fig. 1*A*,*D*). Spike-and-wave epileptiform discharges comprised a spike followed by a slow wave in the $\delta$ range with a peak frequency ∼3 Hz (Fig. 1*B*,*E*). The unit spike burst occurred at the LFP spike's downward deflection (Fig. 1*G–I*). In contrast, the following upward $\delta$ wave accompanied the cessation of unit spikes (Fig. 1*K*). In the mixed pattern of epileptiform discharges, LFP epileptiform spikes appeared with various values of frequency power (Fig. 1*C*,*F*), whereas bursts of unit spikes fired during the downward deflections of epileptiform spikes. Notably, the unit spike firing decreased at the following upward slow wave on LFPs. These results indicate that the unit activity during seizures is highly coupled to the epileptiform discharges observed on LFPs (Fig. 1*J–L*). This finding leads to the hypothesis that epileptic LFPs can be modulated by altering unit spike activity. To investigate this, we examine unit spikes, LFP oscillations and their coupling in response to neuromodulation by ctDCS.

### Cathodal weak direct current stimulation reduced polyspike epileptiform discharges and enhanced $\delta$ oscillations in LFPs

The EF amplitude at CA1 was correlated with the current intensity applied via the tDCS epicranial cannula electrode. A current of 1 mA induced an EF amplitude of 0.9 mV/mm at CA1 (Fig. 2*A*,*B*). To evaluate population-level epileptic excitability changes by ctDCS, we analysed the amplitudes and frequencies of the epileptic spikes in CA1 LFPs after KA induction. The post-KA epileptiform discharges collected under anaesthesia evolved to electrographic status epilepticus, with sustained polyspikes lasting over 10 min before stimulation (Hirsch et al., 2021). The KA-induced polyspikes were labelled, showing increased 25–50 Hz $\gamma$ power, and calculated across the pre-stimulation, stimulation and post-stimulation periods (Fig. 2*C*; Wu et al., 2020). When the stimulation effect over

time within group was examined, the epileptic spike counts significantly decreased after ctDCS at P60–90 compared to P0–30 (one-way ANOVA, $F_{(3, 37)} = 4.24$, $p = 0.011$; *post hoc* Tukey's multiple comparisons test, $p = 0.006$), but this reduction was not observed in the sham-stimulation group ($F_{(3, 36)} = 2.60$, $p = 0.067$). The changes in LFP epileptic spike numbers normalized to pre-stimulation levels were compared between the ctDCS and sham groups during and after the stimulation periods. Outliers with values exceeding the mean plus two SDs were excluded from the analysis. Compared to sham stimulation, the epileptic spikes in the ctDCS group decreased during stimulation (unpaired *t* test, $p = 0.024$) and at 60–90 min post-stimulation ($p = 0.019$, tDCS: 11 rats, sham: 11 rats; Fig. 2D). When the spike amplitude AUCs during stimulation (stim) and post-stim phases was compared to pre-stim levels in the ctDCS and sham groups, we observed a significant decrease in LFP spike amplitude after ctDCS lasting up to 90 min post-stimulation (Kruskal–Wallis test, stim-period $p < 0.0001$; *post hoc* Dunn's multiple comparisons test, when compared to pre-stim, $p = 0.004$ at P0–30,

$p = 0.002$ at P30–60 and $p < 0.0001$ at P60–90; compared to stim, $p = 0.005$ at P60–90; tDCS: 11 rats; Fig. 2E,F). This reduction was not significant in sham-treated rats (Kruskal–Wallis test, $p = 0.051$; sham: 8 rats; Fig. 2G,H). We compared low-frequency power changes in PSD during pre-stim, stim and post-stim periods within groups. A significant increase in $\delta$ power ($\geq 0.5$ to $< 4$ Hz) was observed in the ctDCS group during stimulation and 0–30 min after stimulation compared to pre-stim levels, followed by a subsequent decline. $\delta$ power at 30–90 min after stimulation returned to baseline levels (two-way ANOVA, stim-period $F_{(4, 20,808)} = 8.471$, $p < 0.0001$, frequency $F_{(50, 20,808)} = 47.87$, $p < 0.0001$, interaction $F_{(200, 20,808)} = 1.982$, $p < 0.0001$, tDCS: 14 rats; Fig. 2I). In contrast, $\delta$ power decreased in seizure animals after sham stimulation (stim-periods $F_{(4, 15,198)} = 9.780$, $p < 0.0001$, frequency $F_{(50, 15,198)} = 277.1$, $p < 0.0001$, interaction $F_{(200, 15,198)} = 3.460$, $p < 0.0001$, sham: 14 rats; Fig. 2J). The pre-stim baseline did not differ significantly between the ctDCS and sham groups in the frequency range of 0.5–5 Hz (two-way ANOVA, $F_{(1, 888)} = 0.075$, $p = 0.785$) and 0.5–50 Hz ($p = 0.996$; Fig. 2K). $\delta$ power ($\geq 0.5$ to $< 4$

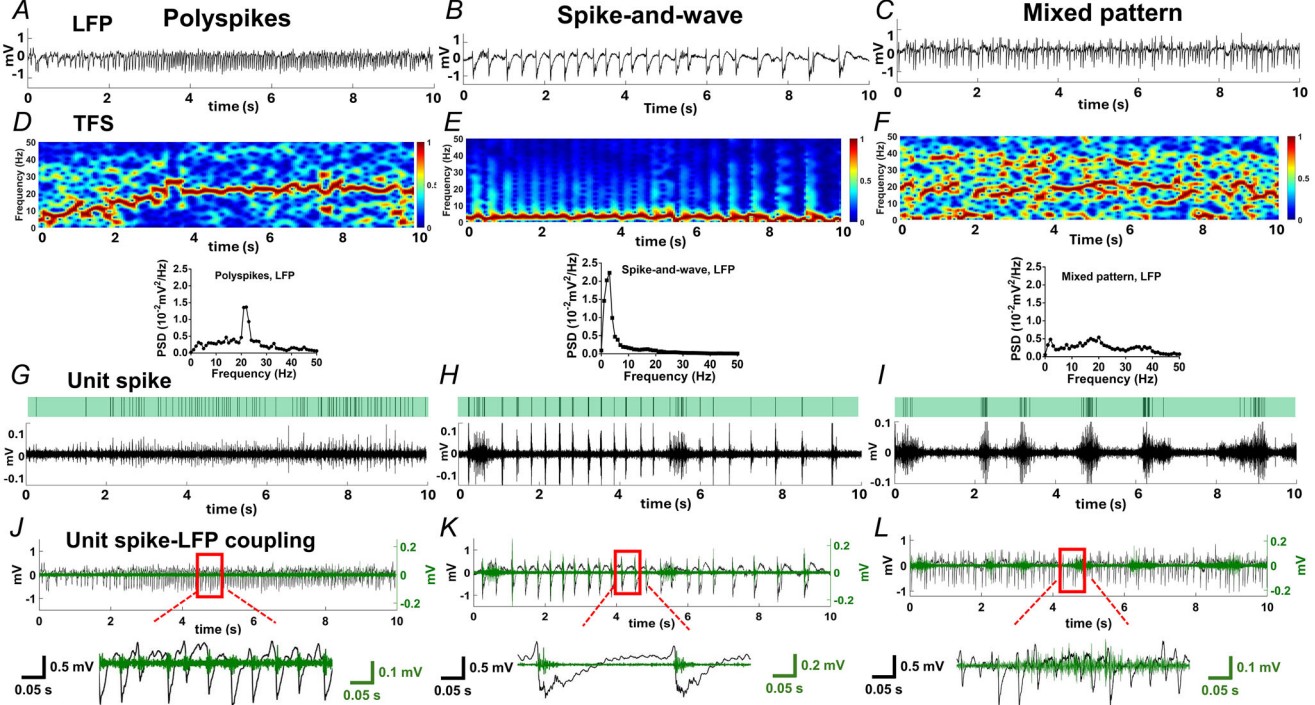

**Figure 1. Neuronal unit spikes coupled with epileptiform discharges in LFPs**
*A–C*, LFP traces with features of epileptiform discharges recorded at CA1 in a rat with KA (kainic acid)-induced seizures. *A*, Polyspike, *B*, spike-and-wave and *C*, mixed pattern. *D–F*, TFS and PSD corresponding to the LFPs in *A–C*, respectively. *G–I*, Simultaneous recording of unit spikes for *A–C*, respectively. Upper panel, identified unit spikes labelled in the green raster; lower panel, signals of unit spike recording. *J–L*, LFP and unit spike coupling for *A*, *B* and *C*, respectively. *J*, Unit spike–LFP coupling in the polyspike waveform. *K*, Unit spike–LFP coupling in the spike-and-wave waveform. *L*, Unit spike–LFP coupling in a mixed pattern. LFP and unit spike signals are shown in black and green, respectively, with magnified traces. LFP, local field potential; PSD, power spectral density; TFS, time–frequency spectrogram. [Colour figure can be viewed at wileyonlinelibrary.com]

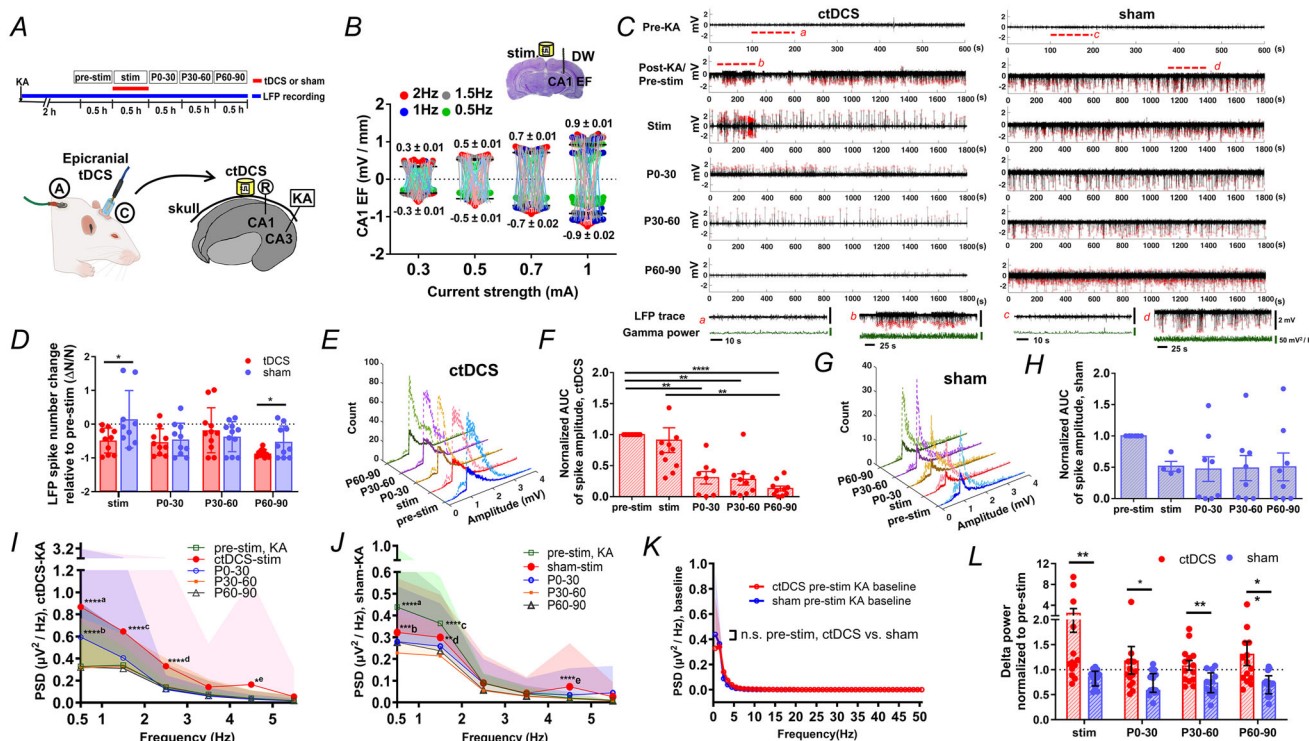

**Figure 2. Reduced epileptic spike number and amplitude, and enhanced δ power in LFPs after cathodal weak direct current stimulation in rats with kainic acid-induced seizures**

*A*, Experimental timeline and assembly of epicranial tDCS electrode, Ⓐ anodal and Ⓒ cathodal stimulation electrodes. Locations of the recording electrode Ⓡ and KA injection. *B*, CA1 EF induced by the applied transcranial currents at varying strengths. Right upper schematic figure showing the epicranial stimulation cannula and DW electrodes for CA1 EF measurements. *C*, Representative LFPs of pre-stim, stim and post-stim periods in ctDCS- (left) and sham-stimulated (right) rats. Enlarged traces of polyspike epileptiform discharges labelled as epileptic spikes with red dots, along with the corresponding 25–50 Hz γ power sum shown at the bottom. *D*, Comparison of LFP epileptic spike numbers individually normalized to pre-stim in ctDCS- and sham-treated rats during and after stimulations. *E*, Amplitude distributions of the identified LFP spikes in ctDCS-treated seizure rats. *F*, Comparison of the AUCs of spike amplitudes normalized to pre-stim individually across time in ctDCS-treated seizure rats ($n = 11$). *G*, Amplitude distributions of the identified spikes in sham-treated rats. *H*, Comparison of normalized AUCs of spike amplitudes across time in sham-treated seizure rats ($n = 8$). *I*, Comparison of PSD for the pre-stim, stim and post-stim periods in ctDCS-treated rats. ****In a, ≥0.5 to <1 Hz, ctDCS-stim compared with pre-stim, P0–30, P30–60 and P60–90, $p < 0.0001$. ****In b, ≥0.5 to <1 Hz, P0–30 compared with P30–60 and P60–90, $p < 0.0001$. ****In c, ≥1 to <2 Hz, ctDCS-stim compared with pre-stim, P0–30, P30–60 and P60–90, $p < 0.0001$. ****In d, ≥2 to <3 Hz, ctDCS-stim compared with pre-stim, P0–30, P30–60 and P60–90, $p < 0.0001$. *In e, ≥4 to <5 Hz, ctDCS-stim compared with pre-stim ($p = 0.027$), P0–30 ($p = 0.014$), P30–60 ($p = 0.024$) and P60–90 ($p = 0.019$), $p < 0.05$; $n = 14$. *J*, Comparison of PSD for the pre-stim, stim and post-stim periods in sham-stimulation groups. ****In a, ≥0.5 to <1 Hz, pre-stim compared with sham-stim, P0–30, P30–60 and P60–90, $p < 0.0001$; sham-stim compared with P30–60, $p < 0.0001$; P0–30 compared with P30–60, $p < 0.0001$. ***In b, ≥0.5 to <1 Hz, sham-stim compared with P0–30 and P60–90, $p < 0.001$; P30–60 compared with P60–90, $p < 0.001$. ****In c, ≥1 to <2 Hz, pre-stim compared with sham-stim, P0–30, P30–60 and P60–90, $p < 0.0001$; sham-stim compared with P30–60 and P60–90, $p < 0.0001$. **In d, ≥1 to <2 Hz, sham-stim compared with P0–30, $p = 0.001$. ****In e, ≥4 to <5 Hz, sham-stim compared with pre-stim, P30–60, P60–90, $p < 0.0001$; $n = 14$. *K*, Comparing the baseline PSD of post-KA-induced hippocampal seizures in ctDCS ($n = 14$) and sham groups ($n = 14$). *L*, Comparing δ power (≥0.5 to <4 Hz) normalized to the pre-stim power in each rat between ctDCS and sham groups during stim, P0–30, P30–60, P60–90 ($n = 14$ in each group). *$p < 0.05$, **$p < 0.01$, ***$p < 0.001$ and ****$p < 0.0001$. *n*, animal number. Solid colour curves with lighter dashed lines, mean with SD in *E* and *G*. Colour curves with shaded bands, mean with SD in *I–K*. Bars, median ± IQR (interquartile range) in *F*, *H* and *L*. AUC, area under the curve; ctDCS, cathodal transcranial direct current stimulation; DW, double wire; EF, electric field; KA, kainic acid; LFP, local field potential, n.s., non-significant; PSD, power spectral density; pre-stim, pre-stimulation; stim, stimulation, P0–30, 0–30 min; P30–60, 30–60 min and P60–90, 60–90 min after stimulation. [Colour figure can be viewed at wileyonlinelibrary.com]

Hz), normalized to pre-stim level, was significantly higher in the ctDCS group during stimulation ($p = 0.001$), 0–30 min ($p = 0.024$), 30–60 min ($p = 0.003$) and 60–90 min post-stimulation ($p = 0.002$) compared to the sham group (Mann–Whitney test, tDCS: 14 rats, sham: 14 rats; Fig. 2*L*). No significant differences were observed in $\theta$ ($\geq 4$ to $< 8$ Hz), $\alpha$ ($\geq 8$ to $< 12$ Hz), $\beta$ ($\geq 12$ to $< 25$ Hz) and $\gamma$ ($\geq 25$ to $< 50$ Hz) band power between the ctDCS and sham groups (Fig. A1).

### Cathodal weak direct current stimulation reduced CA1 excitatory neuronal firing in rats with KA-induced hippocampal seizures

Because neuronal unit firings were correlated with the epileptiform discharges in LFPs and $\delta$ power was enhanced after ctDCS, we tested the hypothesis that ctDCS changes neuronal spike firings. The unit spikes of putative excitatory neurons identified at CA1 stratum pyramidale from animals with KA-induced hippocampal seizures revealed distinct waveform and firing frequency (mean value, peak-to-peak spike width: 0.75 ms, amplitude: 54 μV, left descending slope: $-0.15$ mV/ms, right ascending slope: 0.13 mV/ms, 5.48 Hz; Fig. 3*A*). The basal firing frequency of unit spikes significantly increased in rats with KA-induced seizures (12 rats) compared to saline-treated controls (6 rats), indicating heightened excitatory neuronal activity during seizures (Mann–Whitney test, $p < 0.0001$; Fig. 3*B*). tDCS changed unit spikes in a polarity-specific manner with current strength dosing responses. The unit spike number was significantly decreased by cathodal stimulation and increased by anodal stimulation (atDCS) at 1 mA rather than 0.5 mA in rats with KA-induced seizures (two-way ANOVA, 1 mA-atDCS *vs.* 1 mA-ctDCS, $F_{(1, 753)} = 202.9$, $p < 0.0001$; 1 mA-atDCS *vs.* 0.5 mA-atDCS, $F_{(1, 186)} = 52.52$, $p < 0.0001$; 1 mA-ctDCS *vs.* 0.5 mA-ctDCS, $F_{(1, 755)} = 21.02$, $p < 0.0001$; 0.5 mA-atDCS *vs.* 0.5 mA-ctDCS, $F_{(1, 188)} = 0.26$, $p = 0.61$; Fig. 3*C*). The unit spike number changes relative to the pre-stim spikes was significantly reduced by ctDCS (14 rats) compared to sham stimulation (14 rats), particularly at P30–60 and P60–90 (two-way ANOVA, $F_{(1, 529)} = 108.9$, $p < 0.0001$; *post hoc* Bonferroni multiple comparisons test, $p = 0.001$ at stim, $p = 0.005$ at P0–30, $p < 0.0001$ at P30–60, $p < 0.0001$ at P60–90; Fig. 3*D* and Fig. A2). The increased unit spikes in sham stimulation reflected the ongoing epileptic excitation, whereas ctDCS significantly reduced it. In the ctDCS group, the unit spike frequency reduced over time from pre-stim to P60–90 after ctDCS (mean value 5.48 to 3.4, median 2.8 to 0.77 spikes/s, Kruskal–Wallis test, $p = 0.042$, tDCS: 14 rats; Fig. 3*E*), whereas it increased after sham stimulation (Kruskal–Wallis test, $p = 0.006$;

sham: 14 rats; Fig. 3*F*). The reduction in unit spikes by ctDCS was also observed in the naive control rats during ctDCS treatment (Fig. A3). The neuronal type of the recorded unit spikes was validated using optogenetic inhibition approaches in separated experiments. The unit spikes in similar waveforms and locations as the putative excitatory neurons were analysed for their responses to optogenetic inhibition of CaMKII-expressed excitatory neurons and VGAT-cre-expressed GABAergic inhibitory neurons, respectively. The unit spike number normalized to the mean value of pre-optostim was reduced when CaMKII-expressed neurons were optogenetically inhibited compared to those at pre-optostim ($p < 0.0001$) and post-optostim (Mann–Whitney test, $p < 0.0001$, three mice; Fig. 3*G*). In contrast, the unit spike number increased when VGAT-expressed GABAergic neurons were optogenetically inhibited compared to those at pre-optostim ($p < 0.0001$) and post-optostim (Mann–Whitney test, $p = 0.001$, three mice; Fig. 3*H*). These data suggest that the recorded unit spikes are possibly generated by excitatory CaMKII neurons and receive inhibitory inputs from GABAergic interneurons.

### Cathodal weak direct current stimulation increased coupling strength between unit spikes and $\delta$ waves

Since the unit spike number decreased and the $\delta$ power increased after ctDCS, we explored the hypothesis that the reduced neuronal unit spike entrains LFPs into the $\delta$-ranged oscillation by the coupling between unit spikes and $\delta$ waves. A representative CA1 recording from LFPs and neuronal unit spikes showed the changes during and after ctDCS. Unit spike frequency was decreased during and after ctDCS, and the low frequency power was increased by ctDCS at LFP (Fig. 4*A–F*). Compared to pre-stim baseline, the coupling strength of unit spikes to $\delta$ waves was increased during stimulation (Mann–Whitney test, $p = 0.006$, tDCS: 14 rats) and at P60–90 ($p = 0.028$) in ctDCS-treated rats (unit spike numbers: pre-stim: 53,454; stim: 39,630; P0–30: 43,353; P30–60: 39,149; P60–90: 38,239; Fig. 4*G,H*). This increase was not observed in the sham group, showing a reduced coupling strength of the unit spike and $\delta$ wave at P60–90 compared to pre-stim (Mann–Whitney test, $p = 0.014$, sham: 14 rats), stim ($p = 0.0009$) and P30-60 ($p = 0.012$; unit spike numbers: pre-stim: 49,976; stim: 61,189; P0–30: 61,673; P30–60: 88,304; P60–90: 94,577; Fig. 4*I*). Although ctDCS increased the coupling strength between unit spikes and $\delta$ waves, it did not cause a concentration of unit spike firing phase. Both the ctDCS and sham groups exhibited no strongly preferred firing phases on the distribution of unit spikes to LFP phases (mean phase: ctDCS, pre-stim $-4.54°$, stim 2.62°, P60–90, 0.5°; sham, pre-stim $-2.62°$, stim $-1.44°$, P60–90, 0.92°; average resultant vector length is $< 0.45$ in Watson–Williams test; Fig. 4*J,K*).

## Inhibitory short-term plasticity changes induced by cathodal weak direct current stimulation in rats with hippocampal seizures

We next studied the possibility of ctDCS-induced plasticity changes corresponding to the after-effect of stimulations. The *in vivo* P2/P1 ratio at different ISIs was calculated at CA1 before (pre), immediately after (post), 60 min after (P60) and 90 min after (P90) the stimulation (Fig. 5*A*). When P2/P1 ratio normalized to pre-stim baseline was compared between ctDCS (nine rats) and sham (four rats) treated KA rats, ctDCS-treated

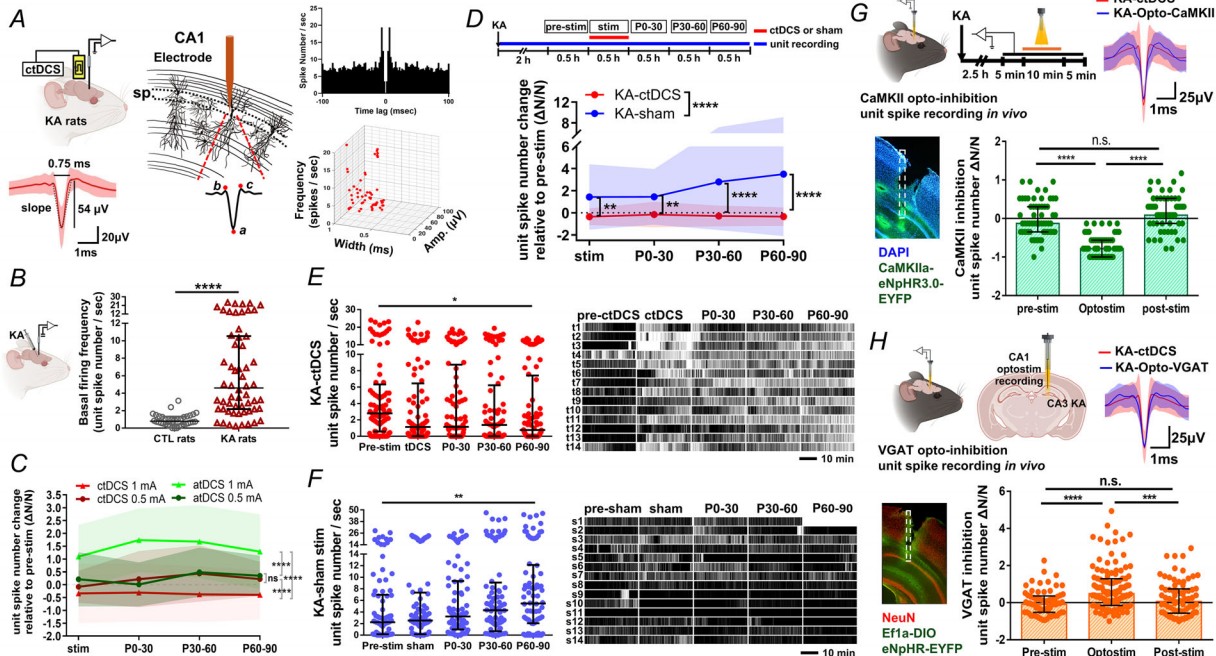

**Figure 3. Decreased excitatory neuronal firing by cathodal weak direct current stimulation in rats with KA-induced seizures**

*A*, *In vivo* set-up (left upper panel) and schematic location of the unit spike recording electrode at CA1 stratum pyramidale. *a*, The trough; *b* and *c*, two peaks of the spike; the distance between *b* and *c* on the *x*-axis represents the peak-to-peak spike width; the distance between *a* and the higher peak on the *y*-axis indicates the spike amplitude (middle panel). Waveform presented with mean values of the spike width, amplitude and slope (left lower panel). Autocorrelation and spike feature distribution based on spike width, amplitude and firing frequency (right panel). *B*, *In vivo* set-up and comparison of the basal firing frequencies of the unit spikes between rats with KA-induced seizures ($n = 12$) and naive controls ($n = 6$). *C*, Comparison of unit spike number changes between anodal tDCS 0.5 mA ($n = 3$), anodal tDCS 1 mA ($n = 3$), cathodal tDCS 0.5 mA ($n = 3$) and cathodal tDCS 1 mA ($n = 14$) in KA-induced seizure rats. *D*, Experimental timeline and the comparison of the changes in unit spike numbers between ctDCS-treated and sham-treated KA rats over time. *E*, Comparison of unit spike frequencies across time in ctDCS-treated KA rats ($n = 14$, left panel). Unit spike raster across pre-ctDCS, ctDCS and post-ctDCS periods in all animals (t1–t14, right panel). *F*, Comparison of unit spike frequencies across time in sham-stimulated KA rats ($n = 14$, left panel). Unit spike raster across pre-sham, sham and post-sham periods in all animals (s1–s14, right panel). P60–90 was not recorded in animal s1. *G*, Upper panel, *in vivo* set-up and experimental timeline. Aligned unit spike waveforms detected from animals with KA-induced seizures treated with CaMKII opto-inhibition and ctDCS. Left lower panel, viral expression in CA1 injected with pAAV-CaMKIIa-eNpHR3.0-EYFP, shown in immunofluorescent staining of DAPI (blue) and EYFP virus (green). Right lower panel, comparison of the unit spike number changes at pre-optostim, optostim and post-optostim periods of optogenetic inhibition to CaMKIIa-expressed excitatory neurons. *H*, Upper panel, *in vivo* set-up and schematic location of optofibre and unit recording electrode in mice with KA-induced seizure. Aligned unit spike waveforms detected from animals with KA-induced seizures treated with VGAT opto-inhibition and ctDCS. Left lower panel, viral expression in VGAT-cre mice injected with pAAV-Ef1a-DIO eNpHR-EYFP, shown in immunofluorescent staining of NeuN (red) and EYFP virus (green). Right lower panel, comparison of the unit spike number changes at pre-optostim, optostim and post-optostim periods of optogenetic inhibition to VGAT-Cre-expressed GABAergic inhibitory neurons. *$p < 0.05$, **$p < 0.01$, ***$p < 0.001$ and ****$p < 0.0001$. *n*, animal number. Colour curves with bands, mean ± SD in *C* and *D*. Bars, median ± IQR (interquartile range) in *B* and *E–H*. amp, amplitude; ctDCS, cathodal transcranial direct current stimulation; CTL, naive control rats; KA, kainic acid; sp, stratum pyramidale; stim, stimulation; P0–30, 0–30 min; P30–60, 30–60 min; and P60–90, 60–90 min after stimulation. [Colour figure can be viewed at wileyonlinelibrary.com]

rats exhibited a reduction at ISI 5 ms (two-way ANOVA, tDCS $F_{(1, 512)} = 30.10$, $p < 0.0001$; period $F_{(3, 512)} = 4.002$, $p = 0.008$; interaction $F_{(3, 512)} = 4.949$, $p = 0.002$; *post hoc*, $p < 0.0001$ at post-stim, $p = 0.002$ at P90; Fig. 5*B*), ISI 20 ms (ctDCS $F_{(1, 512)} = 52.26$, $p < 0.0001$, *post hoc*, $p < 0.0001$ at post-stim, $p < 0.0001$ at P60, $p = 0.002$ at P90; Fig. 5*C,D*) and ISI 100 ms ($F_{(1, 512)} = 29.59$, $p < 0.0001$; Fig. A4). The reduction occurred immediately after stimulation and lasted for 90 min, exhibiting an inhibitory short-term plasticity change after stimulation. P2/P1 showed no significant difference between the ctDCS and sham groups at ISIs of 3.33, 10 and 40 ms (Fig. A4).

## Decreased CaMKII⁺ neuronal activation and BDNF expressions while increased GAD⁺ neuronal activation after cathodal weak direct current stimulation in rats with KA-induced seizures

Since seizure excitability was decreased in neuronal unit spikes, we studied whether the reduction was related to the changes in inhibitory and excitatory neuronal activations. Hippocampal neuronal activation was evaluated using c-Fos in CaMKII- and GAD-labelled excitatory and inhibitory cells, respectively, at the dorsal hippocampal CA1 and DG below the stimulation cannula and the electrode recording sites (Fig. 6*A*).

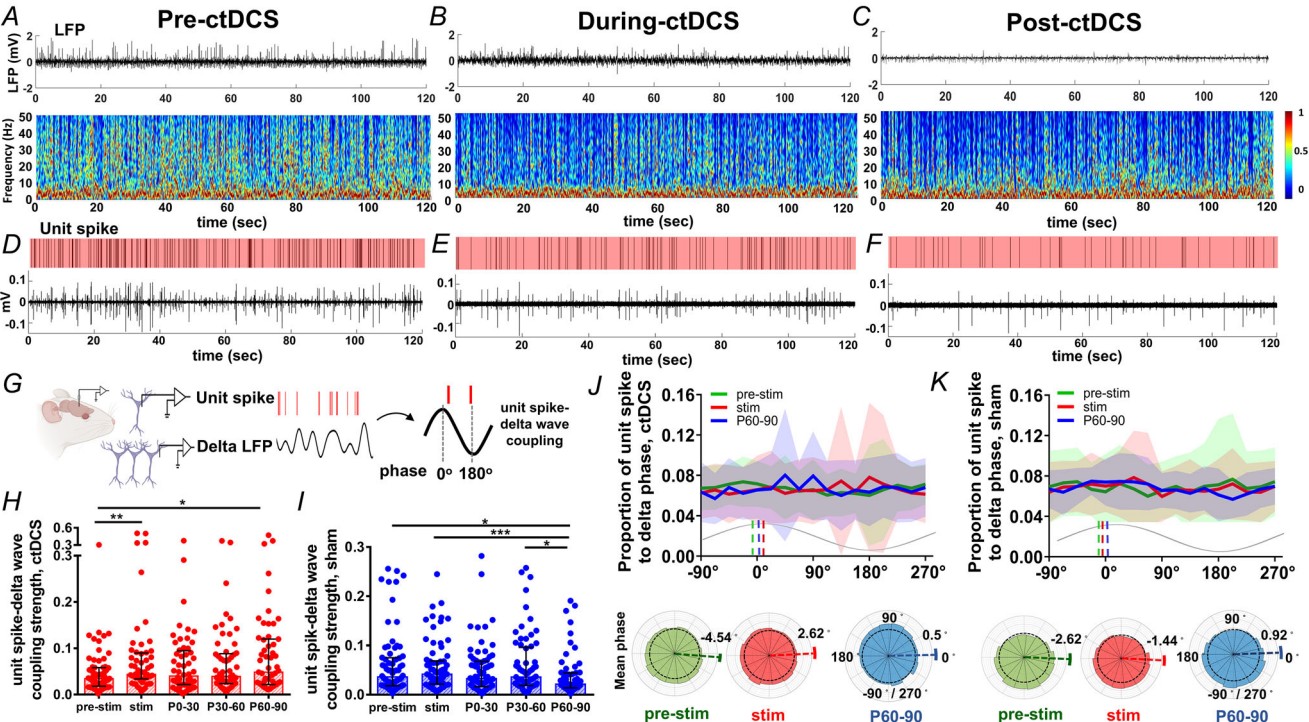

**Figure 4. Enhanced coupling strength between unit spikes and δ waves after cathodal weak direct current stimulation in rats with KA-induced seizures**

*A–C*, Representative recordings of LFPs and corresponding TFS in the pre-stim (*A*), stim (*B*) and post-stim (*C*) periods in a ctDCS-treated rat with KA-induced hippocampal seizures. *D–F*, Simultaneous extracellular unit spike recording and the sorted unit spikes labelled in red raster, corresponding to *A–C*. *G*, Illustration of unit spike–δ wave coupling from *in vivo* LFP and unit spike recording. *H*, Coupling strength of the unit spikes and δ waves in ctDCS-treated rats (*n* = 14) at pre-stim, stim, P0–30, P30–60 and P60–90. *I*, Coupling strength of the unit spikes and δ waves in sham-treated rats (*n* = 14) in the pre-stim, stim, P0–30, P30–60 and P60–90. *J*, Upper panel, proportion of the unit spikes to δ phases in ctDCS-treated rats, with mean phase shown in degrees and indicated by dashed lines. Lower panel, circular histogram illustrating the firing phase across all unit spikes. Dashed coloured lines represent the mean phase, whereas the coloured angular lines correspond to one SD across unit spikes. *K*, Upper panel, proportion of the unit spikes to δ phases in sham-treated rats. Lower panel, circular histogram illustrating the firing phase across all unit spikes. *\*p < 0.05*, *\*\*p < 0.01* and *\*\*\*p < 0.001*. *n*, animal number. Bars, median ± IQR (interquartile range) in *H* and *J*. Colour curves with bands, mean ± SD in *J* and *K*. ctDCS, cathodal transcranial direct current stimulation; LFP, local field potential; stim, stimulation; TFS, time–frequency spectrogram; stim, stimulation; P0–30, 0–30 min; P30–60, 30–60 min; and P60–90, 60–90 min after stimulation. [Colour figure can be viewed at wileyonlinelibrary.com]

BDNF protein expressions decreased in ctDCS (six rats) compared to sham (six rats) treated animals (Mann–Whitney $U$ test, $p = 0.002$; Fig. 6B). c-Fos intensity was significantly decreased in ctDCS-treated rats (six rats) compared to sham-treated rats (six rats) at both stimulation ($p < 0.0001$) and recording electrode sites ($p < 0.0001$). The decrease in c-Fos intensity by ctDCS at the stimulation site exceeded that at the recording site ($p < 0.0001$; Fig. 6C), indicating a spatial variation with a greater extent below the stimulation site and lesser at the adjacent site. At the stimulation site, c-Fos activation in CaMKII$^+$ neurons at CA1 pyramidal cell layer and DG granular cell layer was significantly reduced in ctDCS-treated rats compared to sham-treated rats (Mann–Whitney $U$ test, $p = 0.003$, tDCS: 6 rats, sham: 6 rats), with a trend towards reduction observed at the recording site ($p = 0.06$; Fig. 6D). Conversely, c-Fos activation in GAD$^+$ neurons exhibited an increase in ctDCS-treated rats versus sham-treated rats at the stimulation site ($p = 0.04$), with a tendency towards an increase at the recording site ($p = 0.24$; Fig. 6E). These results indicate that the inhibitory effect of ctDCS on seizures was mediated by reducing activation in CaMKII-labelled excitatory neurons and enhancing activation in GAD-labelled GABAergic inhibitory neurons, exhibiting cell type-specific responses to ctDCS.

## Discussion

Our investigation revealed different patterns of epileptiform discharges in LFPs and the coupling of unit spike bursts and epileptiform discharges at the downward spike deflection. This coupling strongly suggests that the depolarized action potentials of neurons contribute to the generation of epileptiform spikes in LFP. $\delta$ waves consist of quasi-periodic potential shifts with an upward slow wave, consistent with a disfacilitatory event and a pause in spike firing (Buzsaki et al., 2012). However,

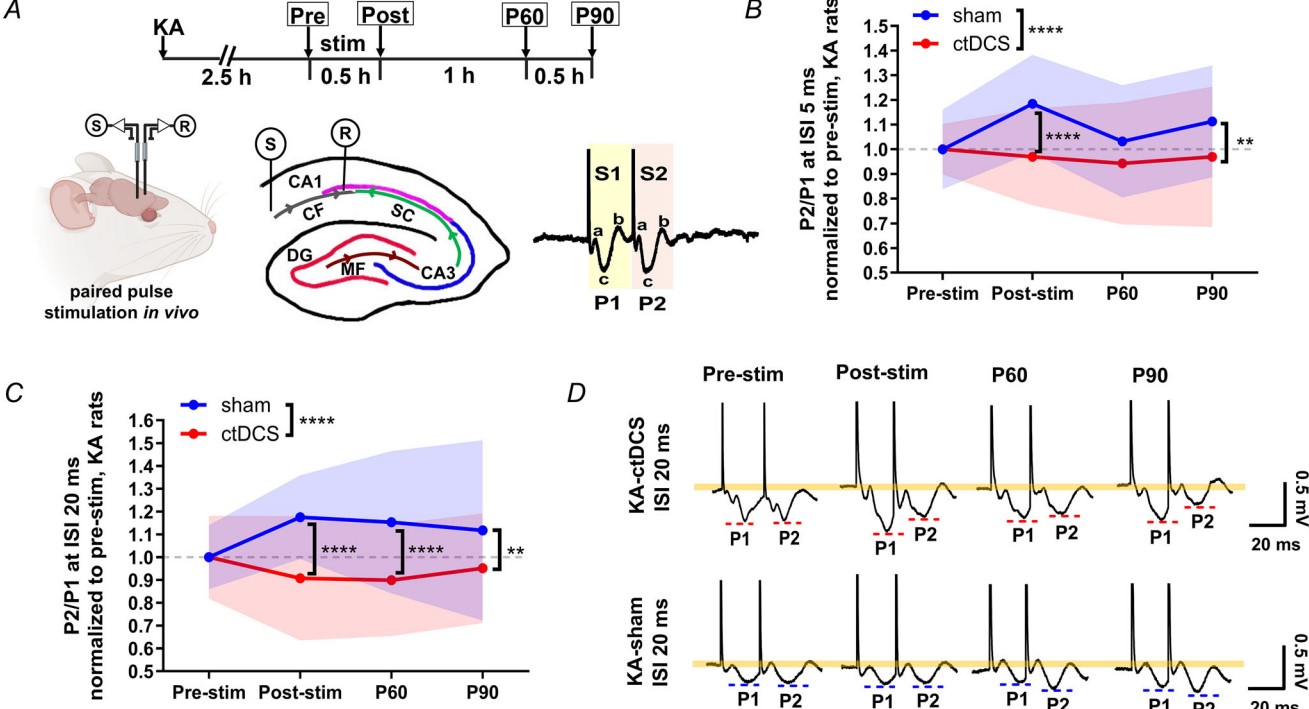

**Figure 5. Inhibitory short-term plasticity changes induced by cathodal weak direct current stimulation in rats with hippocampal seizures**
*A*, Experimental timeline, *in vivo* set-up and schematic graph of paired-pulse stimulation site Ⓢ at ventral hippocampal commissure and recording site Ⓡ at CA1. An illustration of PPS (paired-pulse stimulation) responses; P1 indicates the response evoked by the first stimulation pulse S1 and P2 the response evoked by the second stimulation pulse S2. $P = [(a + b)/2] - c$ from y-axis values; *a*, the first peak; *b*, the second peak; *c*, the trough. *B*, Comparing normalized P2/P1 ratios at ISI 5 ms between ctDCS ($n = 9$) and sham ($n = 4$) treated KA rats. *C*, Comparing normalized P2/P1 ratios at ISI 20 ms between ctDCS- and sham-treated KA rats. *D*, Examples of P1 and P2 traces at ISI 20 ms in ctDCS- and sham-treated KA rats. **$p < 0.01$ and ****$p < 0.0001$. *n*, animal number. Colour curve with bands, mean ± SD. ctDCS, cathodal transcranial direct current stimulation; DG, dentate gyrus; ISI, interstimulus interval; KA, kainic acid; MF, mossy fibres; SC, Schaffer collaterals; stim, stimulation. Pre, pre-stimulation; post, immediately after ctDCS; P60, 60 min after ctDCS; P90, 90 min after ctDCS. [Colour figure can be viewed at wileyonlinelibrary.com]

this phase-preferential firing pattern would be disrupted in some epileptic animals, varying across hippocampal regions and frequency phases (Shuman et al., 2020). A non-phase-preferential firing of unit spikes to $\delta$ oscillation was shown at CA1 of the animals with status epilepticus in the study. This non-phase-preferential firing pattern refers that neurons fire throughout the $\delta$ phase but does not necessarily imply a reduction in disfacilitatory periods between firings. Notably, our findings revealed reduced unit spike activity, enhanced $\delta$ power (but not in higher-frequency bands), and strengthened unit spike–$\delta$ wave coupling in ctDCS-treated animals. This strengthened coupling suggests that excitatory neuronal firing is temporally tuned to a slower frequency in $\delta$ rhythm, reflecting an overall decrease in neuronal firing excitability and supporting the disfacilitatory

linkage between unit spikes and $\delta$ waves after ctDCS. Additionally, ctDCS reduced the amplitude and number of epileptic spikes, indicating a concurrent decrease in excitatory postsynaptic potentials (EPSPs) among neuron populations (Biasiucci et al., 2019). This reduction in epileptic spikes observed in LFPs can be attributed to the decreased summation of EPSPs from fewer activated population neurons consistent with the reduction in unit spike firings after ctDCS. Interesting, a polarity change in epileptic spikes within LFPs was observed during and after ctDCS compared to pre-stim (Fig. 2C). Based on current flow measurement (Buzsaki et al., 2012), we speculate that this polarity shift may indicate the involvement of inhibitory events, where a change in current flow from inward to outward in the neuronal population suggests a hyperpolarization shift from

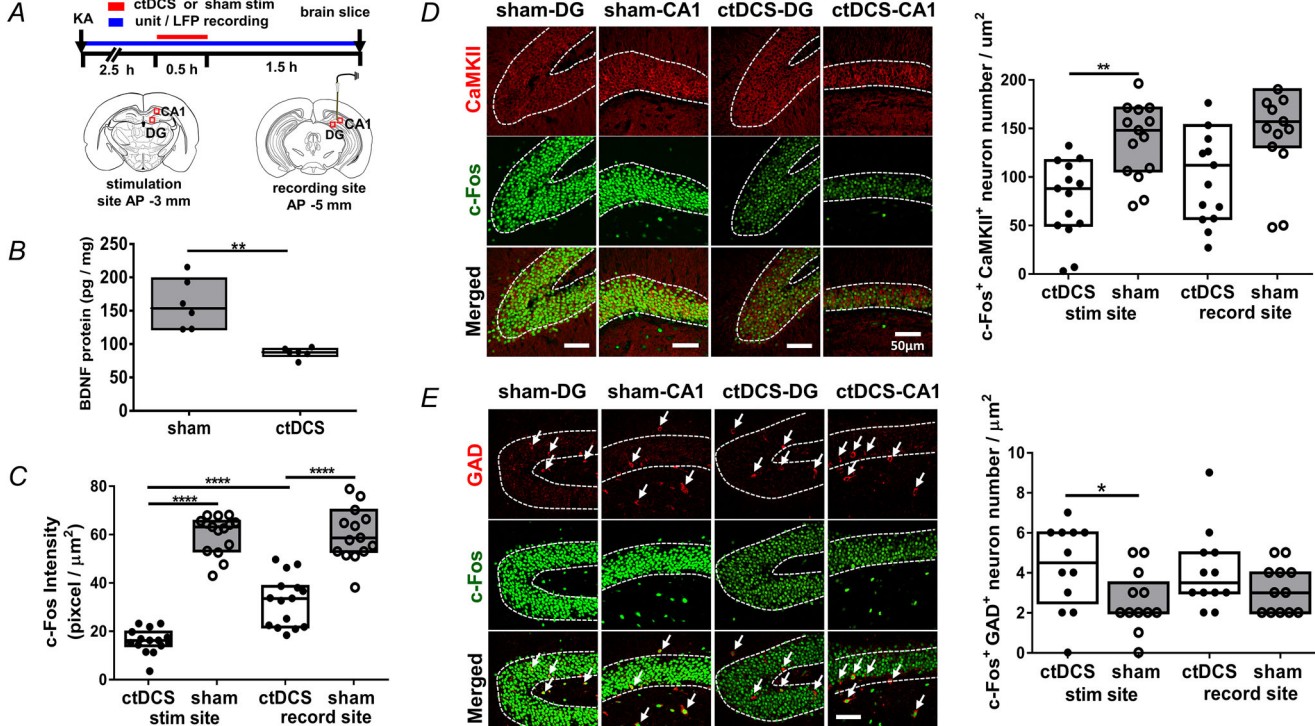

**Figure 6. Changes in BDNF expression and neuronal activation in rats with KA-induced seizures after cathodal weak direct current stimulation**

*A*, Experimental timeline for brain slice harvest after stimulation and recording (upper panel). CA1 and DG (dentate gyrus) regions sampled under the stimulation site and recording site, as indicated by red squares (lower panel). *B*, Comparison of hippocampal BDNF protein expression in ctDCS- ($n = 6$) and sham-treated rats ($n = 6$) with KA-induced hippocampal seizures. *C*, Comparison of ctDCS and sham groups in c-Fos$^+$ intensity at DG and CA1 below the stimulation site and recording site. *D*, Immunofluorescent staining of CaMKII, c-Fos and merged images, white dashed lines indicating the granular cell layer in DG and the pyramidal cell layer in CA1 (left panel). The comparison of ctDCS and sham groups in c-Fos$^+$ CaMKII$^+$ colocalized neuron numbers at CA1 and DG below the stimulation site and recording site (right panel). *E*, Immunofluorescent staining of GAD, c-Fos and merged images, white dashed lines indicating the granular cell layer in DG and the pyramidal cell layer in CA1 (left panel). The comparison of ctDCS and sham in c-Fos$^+$ GAD$^+$ colocalized neuron numbers at CA1 and DG below the stimulation site and recording site (right panel). $^*p < 0.05$, $^{**}p < 0.01$ and $^{****}p < 0.0001$. $n$, animal number. Bar and box, median $\pm$ IQR (interquartile range). BDNF, brain-derived neurotrophic factor; CaMKII, calcium/calmodulin-dependent protein kinase II; ctDCS, cathodal transcranial direct current stimulation; GAD, glutamic acid decarboxylase; KA, kainic acid. [Colour figure can be viewed at wileyonlinelibrary.com]

depolarization. However, this polarity change in LFPs during stimulation was not observed in all animals. Most animals exhibited the same polarity of epileptic spikes in LFPs. Investigating the physiological mechanisms underlying these polarity changes in epileptic field potentials during neuromodulation would be notable for future research. The changes in LFPs, characterized by decreased epileptic spike amplitude and increased $\delta$ oscillations, indicate a reduction in seizure excitability among the population of neurons.

The unit spikes identified at CA1 stratum pyramidale in KA rats are comparable to the putative pyramidal cells based on the waveform but with a higher firing rate (mean value 5.48 Hz) than the basal frequency of pyramidal cells (1.2–2.0 Hz), which could be related to seizures (Csicsvari et al., 1998; Rogers et al., 2021). Neuronal spike firing changes with variability during seizure. The firing frequency in some neurons accelerates twice or thrice during the ictal period than the baseline and decreases by half in the post-ictal period (Alvarado-Rojas et al., 2013; Neumann et al., 2017). Furthermore, these unit spikes decreased when CaMKII$^+$ cells were optically inhibited and increased when VGAT$^+$ cells were inhibited, suggesting that the unit spikes originate from the CaMKII$^+$ principal neurons and receive GABAergic inhibition. The diminution of unit spikes is also consistent with the reduction in neuronal activation measured by c-Fos in CaMKII$^+$-labelled cells at the CA1 pyramidal cell layer after ctDCS. Notably, the decreased unit spikes were aligned with increased $\delta$-range oscillations through the enhancement of unit spike–$\delta$ wave coupling. Neuronal activity has the capacity to modulate brain rhythms by either coupling or decoupling unit spikes with specific LFP oscillations (Tzilivaki et al., 2023). It is possible that different unit spikes synchronize in distinct manners with LFP waves across varying frequency ranges. The unit spike–LFP phase coupling indeed varies between different cell types and is influenced by factors such as cell morphology, perisomatic GABAergic inhibition and the inputs from intrinsic circuitry (Navas-Olive et al., 2020). Although the coupling between unit spikes and $\delta$ oscillations remains elusive in seizure control, our study demonstrates that ctDCS decreases unit spike frequency and enhances LFP $\delta$ oscillations in an *in vivo* seizure animal model, highlighting the increased strength of unit spike–$\delta$ wave coupling in ctDCS-treated rats, linking cellular mechanisms to population-level oscillations.

Understanding how weak current stimulation influences LFP rhythms is crucial given the interest in leveraging non-invasive brain stimulation to modulate brain functions. Our data showed that the applied weak currents change unit spike activities in a polarity-specific manner with dosing effects of current strength, suggesting a broad implication to modulate neuronal activities with

polarity-specific and dosing-dependent weak current stimulations. *In vitro* studies on hippocampal slices showed that weak EFs <1 mV/mm are able to synchronize single neuronal activity (Francis et al., 2003). Our *in vivo* data showed that applying a weak current of 1 mA outside the skull induced an EF of 0.9 mV/mm at the rodent dorsal hippocampal CA1 region, potentially modulating neuronal activity and LFP oscillations during seizures. ctDCS-induced EF interfering with the seizure activity by ephaptic coupling and the hyperpolarized changes in membrane potential caused by cathodal direct currents may underlie the mechanisms (Chang et al., 2015; Subramanian et al., 2022). Moreover the induced EF in our *in vivo* set-up is comparable to other studies but larger than the EF induced by tDCS in the human hippocampus as 0.17 mV/mm (Farahani et al., 2024; Louviot et al., 2022). Variation in the induced EF amplitudes is influenced by factors like brain size, electrode-target distance, neuronal orientation, CSF distribution and tDCS settings.

Our study revealed that ctDCS inhibitory after-effects are associated to short-term synaptic plasticity changes assessed using a PPS algorithm, which is commonly used to assess paired-pulse facilitation (PPF) and paired-pulse depression (PPD). PPF is considered as an increase in presynaptic vesicle release of Ca$^{2+}$ after the second stimulation, whereas PPD indicates a GABA-mediated synaptic inhibition (Davies et al., 1990). In human motor cortex short ISIs of 1.5–5 ms are commonly used to examine short-interval intracortical inhibition, whereas long ISIs of 100–250 ms are used for long-interval intracortical inhibition, through GABAergic inhibition (Cash et al., 2017). Our data indicate that most P1 responses remain stable at P60 and P90 after stimulation, whereas some P1 responses exhibit an enlarged evoked potential immediately post-ctDCS. Meanwhile, the P1 remains consistent in the sham control group. A possible explanation for this P1 potentiation is an increased probability of presynaptic neurotransmitter release and enhanced responsiveness of postsynaptic receptors in certain animals immediately after stimulation. Overall, P2 showed a consistent reduction after ctDCS, and the P2/P1 ratio decreased after ctDCS compared to both the pre-stimulation state and the sham-stimulation group. The increased P2/P1 ratios in the sham-treated rats reflect a facilitated synaptic plasticity in seizure animals, whereas the lasting reduction in P2/P1 for 90 min in the ctDCS group indicates an inhibitory synaptic plasticity change induced after ctDCS. These findings suggest that ctDCS may enhance GABAergic inhibition, with short-term synaptic plasticity changes underpinning ctDCS enduring anti-seizure effects.

BDNF regulates neuronal excitability and synaptic plasticity through tropomyosin receptor kinase B signalling. Clinical studies showed controversial correlations of BDNF levels and epileptogenesis,

confounded by the various subtypes of epilepsy and anti-seizure medication (McGonigal et al., 2023). Hippocampal BDNF levels are correlated with seizure severity and reduced by multisession ctDCS in status epilepticus animal models (Wu et al., 2020). This study showed the decrease in BDNF in animals with hippocampal seizure using single-session ctDCS, indicating BDNF as a biomarker for ctDCS effects in seizure for both conditions. In addition, BDNF reduces the inhibitory postsynaptic currents in PPDs (Wang et al., 2022). The decrease in BDNF, along with the increased activation in GAD$^+$-labelled GABAergic neurons after ctDCS, may explain the changes in inhibitory short-term plasticity observed in our PPS findings. These findings suggest that ctDCS reduces seizure excitability by modulating neuronal activity and synaptic plasticity through reducing CaMKII-mediated excitation and enhancing GABAergic inhibition. It also offers translational insights into the potential electrophysiological biomarkers, such as $\delta$ power, unit spike activity and PPS to monitor ctDCS therapeutic effects in seizure control.

Limitations of this study include the absence of histomorphologic study to validate the neuronal cell types of the identified unit spike. To maintain the unit recording stability during acute seizures, the signals were collected under anaesthesia, which resulted in a lack of behavioural seizures, so we defined epileptic spikes based on electrographic seizures instead. The anaesthesia effects were controlled through the sham-stimulation design in contrast to the ctDCS group. We used single-channel recording instead of tetrodes due to the limited skull space available for implanting the stimulation set-up and recording electrode headstage. Taking advantage of optogenetic manipulation, transgenic mice with KA-induced seizures were used to verify the cell type of the sorted unit spikes. However, the transferability of KA-induced seizures and unit spikes between mice and rats should still be considered. Additionally, although we revealed the relationship between LFPs and unit spikes, a broader spatial recording is necessary to assess the dynamics of the neuron populations contributing to LFPs. Interestingly, the differing responses to ctDCS in GAD$^+$ and CaMKII$^+$-labelled neurons suggest potential variations in cell type-specific susceptibility to ctDCS. Factors like neuronal architecture, orientation to EF and membrane potentials in seizure contexts may influence neuronal responses to ctDCS (Ye et al., 2022). The susceptibility to the weak direct current-induced EFs in different cell types remains to be determined.

## Conclusion

In conclusion, ctDCS significantly decreased the CA1 excitatory neuronal activity while concurrently enhancing $\delta$ oscillation in LFPs in animals with KA-induced hippocampal seizures. The reduced neuronal unit spikes coupled LFPs to the $\delta$-ranged oscillation with an increased coupling strength between unit spikes and $\delta$ waves. Besides, ctDCS instigated an inhibitory short-term plasticity change with lasting effects on neuronal firing and LFP oscillations. The study provides insights into the neuronal mechanisms through which cathodal weak direct current stimulation reduces seizure excitability by modulating neuronal activity and $\delta$ oscillations. These findings hold potential clinical implications for using non-invasive weak direct current stimulation to treat brain disorders characterized by hyperexcitability.

## Appendix

### Appendix methods: epileptic spike detection algorithm

To investigate the acute effects of cathodal direct current stimulation (ctDCS) on epileptic LFPs, a computer-added LFP spike detection and amplitude distribution analysis method is used to calculate the number of epileptic spikes and the amplitude of epileptic spikes corresponding to data from pre-ctDCS, ctDCS and post-ctDCS or sham for comparison. The 30-min signal segments before seizure induction were used as the baseline. The positive and negative peaks after seizure induction were detected. The positive peaks with amplitudes larger than 10 times the average positive baseline and the negative peaks with amplitudes greater than 10 times the negative baseline were retained. The amplitude and the width of each peak were calculated individually to extract the identified epileptic spike. The turning points leading and lagging the peak were employed to be the starting and ending points of the peak, respectively. The average amplitude difference from the peak to the starting point and from the peak to the ending point is defined as the spike amplitude. The width of the spike is the time interval between the starting and ending points. A spike with an absolute amplitude $\geq 1$ mV and a width $\leq 140$ ms was labelled as epileptic spike in LFPs.

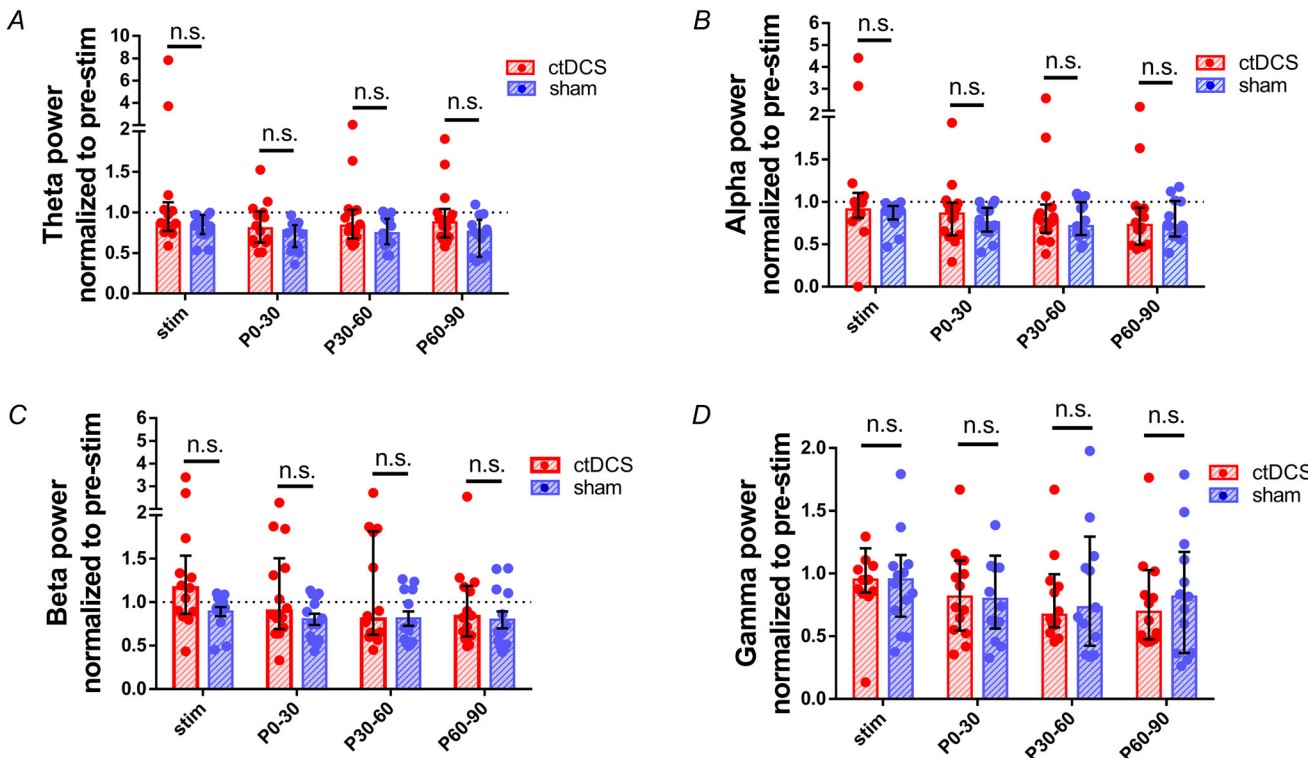

**Figure A1. Frequency-band power comparisons between ctDCS and sham treated rats with KA-induced seizures**

*A*, Comparison of theta power (≥4 to <8 Hz) normalized to the pre-stim power in each rat between ctDCS (*n* = 14) and sham (*n* = 14) groups during stim, P0–30, P30–60, P60–90 (Mann-Whitney test, non-significant difference between ctDCS and sham at each period). *B*, Comparison of normalized alpha power (≥8 to <12 Hz) between ctDCS and sham groups during stim, P0–30, P30–60, P60–90 (Mann-Whitney test, non-significant difference between ctDCS and sham at each period). *C*, Comparison of normalized beta (≥12 to <25 Hz) between ctDCS and sham groups during stim, P0–30, P30–60, P60–90 (Mann-Whitney test, non-significant difference between ctDCS and sham at each period). *D*, Comparison of normalized gamma (≥25 to <50 Hz) between tDCS and sham groups during stim, P0–30, P30–60, P60–90 (Mann-Whitney test, non-significant difference between ctDCS and sham at each period). n, animal number. bars, median ± IQR. n.s., non-significant. ctDCS cathodal transcranial direct current stimulation. [Colour figure can be viewed at wileyonlinelibrary.com]

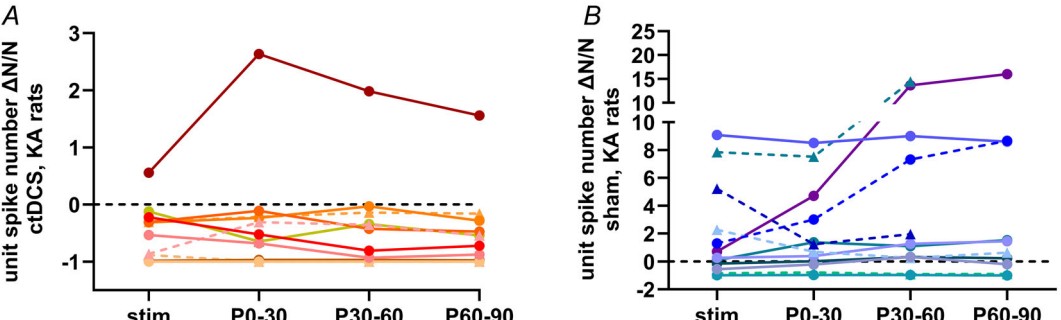

**Figure A2. Changes in unit spike numbers for individual animals treated with tDCS or sham stimulation**

*A*, Unit spike number changes relative to pre-stimulation levels for each KA rat in the ctDCS group, shown across the periods of stim, P0-30, P30-60, and P60-90. *B*, Unit spike number changes relative to pre-stimulation levels for each KA rat in the sham-stimulation group, across the periods of stim, P0-30, P30-60, and P60-90. KA, kainic acid; stim, stimulation; P0-30, 0-30 min post-stimulation; P30-60, 30-60 min post-stimulation; P60-90, 60-90 min post-stimulation. [Colour figure can be viewed at wileyonlinelibrary.com]

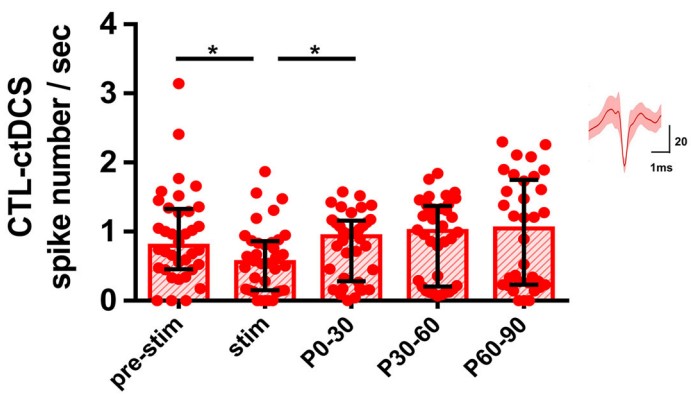

**Figure A3. Comparison of unit spike frequency changes across time in naive control rats treated with ctDCS**
The unit spike frequency was decreased at stim when compared with pre-stim (Mann–Whitney test, $p = 0.023$) and P0-30 ($p = 0.032$) in ctDCS-treated CTL rats ($n = 6$). Unit spike waveform presented at right depiction. n, animal number. bars, median ± IQR. CTL, control; stim, stimulation; P0-30, 0-30 min following stimulation; P30-60, 30-60 min following stimulation; P60-90, 60-90 min following stimulation. *$p < 0.05$. [Colour figure can be viewed at wileyonlinelibrary.com]

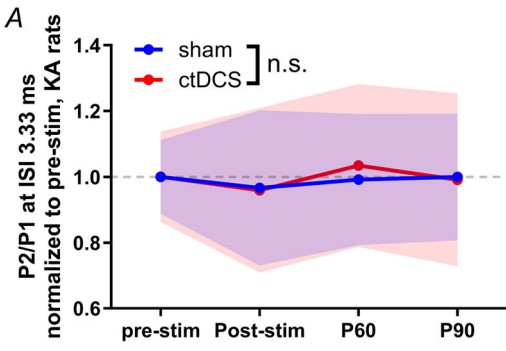

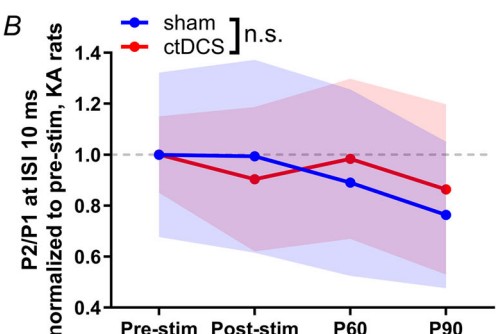

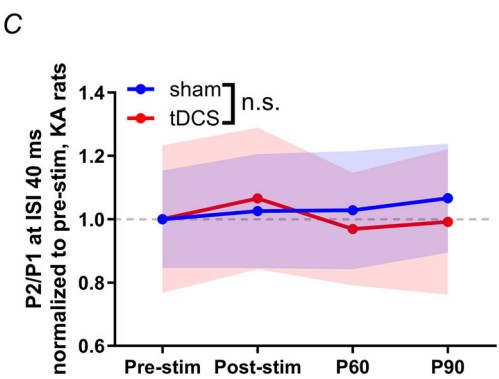

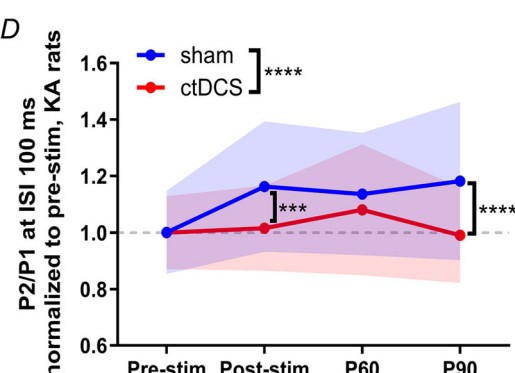

**Figure A4. Comparison of paired-pulse stimulaiton responses at varied interstimulus intervals between ctDCS and sham groups in rats with KA-induced hippocampal seizures**
*A*, Comparing normalized P2/P1 ratios at ISI 3.33 ms between ctDCS ($n = 9$) and sham ($n = 4$) treated KA rats (two-way ANOVA, ctDCS vs. sham, $F_{(1, 512)} = 0.1015$, $P = 0.75$). *B*, Comparing normalized P2/P1 ratios at ISI 10 ms between ctDCS ($n = 9$) and sham ($n = 4$) treated KA rats (two-way ANOVA, ctDCS vs. sham, $F_{(1, 512)} = 0.8369$, $P = 0.36$). *C*, Comparing normalized P2/P1 ratios at ISI 40 ms between ctDCS ($n = 9$) and sham ($n = 4$) treated KA rats (two-way ANOVA, ctDCS vs. sham, $F_{(1, 512)} = 1.463$, $P = 0.22$). *D*, Comparing normalized P2/P1 ratios at ISI 100 ms between ctDCS ($n = 9$) and sham ($n = 4$) treated KA rats (two-way ANOVA, ctDCS vs. sham, $F_{(1, 512)} = 29.59$, $P < 0.0001$; post-hoc, $p < 0.001$ at post-stim, $p < 0.0001$ at P90). n, animal number. bars, mean ± SEM. n.s., non-significant, ***$p < 0.001$ and ****$p < 0.0001$. [Colour figure can be viewed at wileyonlinelibrary.com]

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

## Additional information

### Data availability statement

Data will be made available on request.

### Competing interests

The authors declare no competing interests.

### Author contributions

Y.-J.W., C.-C.C., and D.D. conceived the study. Y.-J.W. designed and supervised the study. C.-C.C., M.-E.C., and Y.-C.H. performed experiments and analyzed data for unit spikes. Y.-J.W., M.-E.C., J.-T.L., and S.-F.L. wrote the spike detection algorithm and analyzed the spikes in LFPs. M.-E.C. performed surgeries, conducted experiments, and analyzed data. Y.-J.W. wrote the manuscript draft. Y.-J.W., K.-S.H., S.-F.L., C.-C.C., and D.D. interpreted the data and revised the manuscript.

### Funding

This work was supported by research grants NSTC 112-2314-B-006-061-MY3, MOST 111-2314-B-006-100 and MOST 110-2628-B-006-027 from the National Science and Technology Council, Taiwan, and NCKUH-11210003 from the National Cheng Kung University Hospital to Dr. Yi-Jen Wu and the National Institutes of Health grants R01NS121084 and R01NS124592 to Dr. Dominique M. Durand.

### Acknowledgements

The authors thank the Bioimaging Core Facility for the support in imaging analysis and the Laboratory Animal Centre, College of Medicine, at the National Cheng Kung University.

### Keywords

brain oscillation, cathodal transcranial direct current stimulation, inhibitory synaptic plasticity, neuromodulation, neuronal excitability, seizure, unit spike–local field potential (LFP) coupling, weak electric field

## Supporting information

Additional supporting information can be found online in the Supporting Information section at the end of the HTML view of the article. Supporting information files available:

**Peer Review History**

