## [Peer Review History · The Journal of Physiology]

Cathodal weak direct current decreases epileptic excitability with reduced neuronal activity and enhanced delta oscillations

Chia-Chu Chiang, Miao-Er Chien, Yu-Chieh Huang, Jyun-Ting Lin, Sheng-Fu Liang, Kuei-Sen Hsu, Dominique M Durand, and Yi-Jen Wu

DOI: 10.1113/JP287969

Corresponding author(s): Yi-Jen Wu (wuyj@mail.ncku.edu.tw)

The following individual(s) involved in review of this submission have agreed to reveal their identity: Wei-Chih Chang (Referee #2)

Review Timeline:

Submission Date:	28-Oct-2024
Editorial Decision:	09-Dec-2024
Revision Received:	08-Feb-2025
Editorial Decision:	20-Feb-2025
Revision Received:	21-Feb-2025
Accepted:	12-Mar-2025

Senior Editor: Katalin Toth

Reviewing Editor: Gareth Morris

Transaction Report:

Dear Dr Wu,

Re: JP-RP-2024-287969 "**Cathodal weak direct current decreases epileptic excitability through reducing neuronal activity and enhancing delta oscillations**" by Chia-Chu Chiang, Miao-Er Chien, Yu-Chieh Huang, Jyun-Ting Lin, Sheng-Fu Liang, Kuei-Sen Hsu, Dominique M Durand, and Yi-Jen Wu

Thank you for submitting your manuscript to The Journal of Physiology. It has been assessed by a Reviewing Editor and by 2 expert referees and we are pleased to tell you that it is potentially acceptable for publication following satisfactory major revision.

REVISION CHECKLIST:

We look forward to receiving your revised submission.

Yours sincerely,

Katalin Toth
Senior Editor
The Journal of Physiology

REQUIRED ITEMS

Missing Abstract figure caption.

- Author photo and profile. First or joint first authors are asked to provide a short biography (no more than 100 words for one author or 150 words in total for joint first authors) and a portrait photograph. These should be uploaded and clearly labelled together in a Word document with the revised version of the manuscript. See Information for Authors for further details.

- Papers must comply with the Statistics Policy: https://jp.msubmit.net/cgi-bin/main.plex?form_type=display_requirements#statistics.

In summary:

- If $n \leq 30$, all data points must be plotted in the figure in a way that reveals their range and distribution. A bar graph with data points overlaid, a box and whisker plot or a violin plot (preferably with data points included) are acceptable formats.

- If $n > 30$, then the entire raw dataset must be made available either as supporting information, or hosted on a not-for-profit repository, e.g. FigShare, with access details provided in the manuscript.

- 'n' clearly defined (e.g. x cells from y slices in z animals) in the Methods. Authors should be mindful of pseudoreplication.

- All relevant 'n' values must be clearly stated in the main text, figures and tables.

- The most appropriate summary statistic (e.g. mean or median and standard deviation) must be used. Standard Error of the Mean (SEM) alone is not permitted.

- Exact p values must be stated. Authors must not use 'greater than' or 'less than'. Exact p values must be stated to three significant figures even when 'no statistical significance' is claimed.

- Please include an Abstract Figure file, as well as the Figure Legend text within the main article file. The Abstract Figure is a piece of artwork designed to give readers an immediate understanding of the research and should summarise the main conclusions. If possible, the image should be easily 'readable' from left to right or top to bottom. It should show the physiological relevance of the manuscript so readers can assess the importance and content of its findings. Abstract Figures should not merely recapitulate other figures in the manuscript. Please try to keep the diagram as simple as possible and without superfluous information that may distract from the main conclusion(s). Abstract Figures must be provided by authors no later than the revised manuscript stage and should be uploaded as a separate file during online submission labelled as File Type 'Abstract Figure'. Please also ensure that you include the figure legend in the main article file. All Abstract Figures should be created using BioRender. Authors should use The Journal's premium BioRender account to export high-resolution images. Details on how to use and access the premium account are included as part of this email.

EDITOR COMMENTS

Reviewing Editor:

Methods Details:

Details of analgesia and anaesthesia are needed in methods section 2.6 (optogenetics). Details of any analgesia are missing in section 2.1. The method of euthanasia needs to be specified for all animals used. Please be sure to add these details in order to meet the Journal's ethical standards.

Ethics Concerns:

No ethical concerns, but please add the details requested above.

Comments for Authors to ensure the paper complies with the Statistics Policy:

Section 2.9 and throughout data analysis - the Journal requires the use of standard deviation rather than SEM - please change this.

Please report exact p values rather than ranges (e.g. line 326 page 14)

Comments to the Author:

Thank you for submitting your manuscript, which has been reviewed by two expert referees. Although both referees express enthusiasm about this work, they offer a number of comments which they feel will improve the strength of the work.

REFEREE COMMENTS

Referee #1:

The manuscript by Chiang et al., investigates the anti-epileptic mechanism of cathodal weak current transcranial stimulation in an acute rat intra-hippocampal kainate model. For this, under anaesthesia, they apply cathodal transcranial direct current and record the LFP as well as unit activity in the hippocampus while the animal is experiencing a prolonged status epilepticus (the recording/stimulation starts after an initial XX hours). They found that following the stimulation, there is a prolonged (for the next 90min) decrease of neuronal firing associated with an increase of delta power as well as a prolonged paired-pulse depression on top of anti-epileptiform effect which is unseen in the sham stimulation group.

Overall, the manuscript is clear and well-illustrated. The data and mechanisms presented are of importance to the epilepsy field to better understand the advantage and limitations of this transcranial stimulation.

Here are my comments that I hope will help consolidate the manuscript:

1. According to the authors and the data, transcranial stimulation would reduce neuronal activity by increasing inhibition since the paired-pulse depression and cfos exp would suggest that there is an increase of inhibition during/after stimulation. It is quite an attractive explanation, and it would also suggest that the stimulation somehow as a differential effect on different cell types (which is very possible). However, the data presented at least in the cfos experiment are a bit thin since the sample size for GABA neurons co-labeled with cfos is about ~2-4 neurons/um² which is very small. The authors mentioned in the methods spike sorting of interneurons as well. It could greatly strength their data if they could analyse the firing pattern of FS interneurons based on the width and ISI. It could help clarify whether some interneurons at least increased their firing during/after the stimulation.

2. The data also suggest that there is a stronger coupling between delta oscillation and neuronal firing in the group receiving the stimulation. The argument being that more delta oscillation more silent firing pause. However, from figure 4i, it seems that there are no preferential oscillation phases of neuronal firing suggesting that there is no more silent period. In addition, are the different units better synchronised between each other?

3. All the statistics on single unit are made as the single unit being the experimental unit. It would also be good to see the data per animal either with a colour code or having in the supp material the same data but within each animal.

4. Figure 2 E and G are unclear to me. The group stim seems to have more spikes and with a bigger amplitude in all the condition compared to the sham one which is going in the opposite direction of the rest especially with the AUC analysis. In addition, what the dashed lines mean is unclear (min and max and the bold traces are the average?).

5. Figure 2 panel K: the power of the low frequencies (\sim delta power) is quite different between the pre-stim sham and treated group with a lower power for the treated group. It does not seem to be the case in panel I and J why? If it is the case that there is a difference, it should be discussed in term of interpretation of the data.
6. Figure 3 panel D: Why the KA-treated group starts below 0 while the sham one starts above? Does it suggest that the decrease observed in the treated group was not already happening before the stim? And the opposite being true for the sham stim? If it is the case how the authors can distinguish a true effect against a drift?
7. Figure 5 D: in the treated group, post-stimulation you see an increase of P1 while P2 stays stable when compared to pre-stimulation. This would not necessarily suggest an paired-pulse depression but more a potentiation of P1.. Can the authors comment on this and eventually discuss it in the manuscript if true.
8. Figure 6: Are the counting made blind to the condition? It should be indicated in the methods section. In addition, was the entire hippocampus analysed or only sub region? If yes, were the region randomly imaged or not?
9. Supp F1: the frequency definition of each power band would be good to add.
10. Methods: Immuno protocol: what is the blocking solution composition. It should be added to the text.
11. Is the anaesthesia during the exp the same as for the surgery? 1.5-2% isoflurane? If yes, the authors could add something to make it clear. If not, what was it?
12. Line 150: it should read ground instead of "grand".

Referee #2:

Chiang, et al., Cathodal weak direct current decreases epileptic excitability through reducing neuronal activity and enhancing delta oscillations

In this study, Chiang et al., applied cathodal direct current stimulation (ctDCS) on an in vivo acute model of seizures, and attempted to uncover the anti-seizing mechanisms of ctDCS, from the aspects of the changes of epileptic spikes, neuronal firing rates, unit-LFP coupling, enhanced inhibition, and c-fos expression. This manuscript provided interesting notions, and the findings may be applicable for clinical usage, but there are still points to clarify:

Major

About the ctDCS. I am sorry if I missed the description, but what were your stimulating patterns? My understanding is: e.g. in Figure 2B, at a current strength of 1 mA, you generated an electrical field from -0.9 to 0.9 mV/mm, and I supposed you switched the polarity (thus you had a positive or a negative field) at 0.5-2 Hz; however, what is the exact length of each polarity? Did you have a short break between each polarity? For how long?

Then, what is the actual pattern of ctDCS used for the rest experiments?

30 minutes of stimulation isn't short; I wonder if your stimuli were good (and stable) for the whole 30 minutes.

Also about the ctDCS itself. Does the ctDCS cause any artifacts? While you made either from 0.5 - 2 Hz ctDCS, I am curious about how the on/off phases of ctDCS affected the LFPs, but I don't see any clues from the recording like Figure 2C.

Different epileptic activities. Chiang et al., described three types of epileptic activities after kainic acid infusion in Results 3.1 and Figure 1; if I did not miss anything important, it seems only 'polyspikes' were used in the following sections. So, I am confused about the need for classification.

Line 302-305, I am not sure I understand the notion.

I think the detected upward events from Figure 2C [Stim] are different from Figure 2C [Post-KA/Pre-stim] or the whole Figure 2C sham group. What was happening? Were these changes in polarities and waveforms common in the ctDCS group? I feel the changes suggest the real ctDCS's effect, and even hint at the involvement of more interneurons (and/or less pyramidal neurons). Maybe you should address this question.

Figure 2D is not very convincing: I would suspect there are still 2 outliers in the sham group at the Stim section; if the two outliers were removed, would they just make the P60-P90 no different?

Figure 2E or G are unclear (the way they are presented), and I cannot grasp anything from the panels. I may not use 'AUC' (AUC is commonly associated with receiver operating characteristic curve) in Figure F and H; what is the reason you wanted to present the area (joules?), but not just the amplitudes? Please specify the purpose.

Unit detection and spike sorting.

It is notoriously difficult to isolate the unit activities from epileptic activities, especially since the authors only had single-channel recordings instead of using tetrodes, which makes it more challenging. One of the important notions in this manuscript is based on the increased/decreased firing rate thus we should be rather careful about false negatives/positives. Using easily changeable measures (i.e. firing frequency) for sorting is risky; would you consider other static measures instead? e.g. descending/ascending slopes, if a trough is following/followed by a positive deflection and then the amplitudes them. Honestly, I felt some firing rates of units are abnormally high (Figure 3B, E, and F; temporally 20-30 Hz might be fine, but continuous 20-30 Hz looks weird) and I could not see a refractory period (~2 ms) from the representative autocorrelation (Figure 3A), thus I worry that couples of units were still not separated.

The raster plots do not reflect the decrease (Figure 3E) and increase (Figure 3F) if the bright bands mean the events.

You may already have a few sorted interneurons, so why not reveal their behaviors? Especially you have mentioned the unit-delta coupling and the potentiated inhibition. On the other hand, I don't feel Figure 3G or 3H is really helpful when the species are different, and I doubt the responses to KA are transferable between species. After spike sorting and checking autocorrelation and cross-correlation, most people would believe your putative pyramidal neurons and interneurons by their mid-width > xxx ms or < xxx ms (Csicsvari et al.).

According to Figure 1G-I or Figure 4D-F, I worry the false negatives were uneven and high from one experiment to another. Were they before or after spike sorting?

Pair-pulse. Figure 5D, this is rather interesting that your P1 changed in the KA-ctDCS rat. As the internal control, P1 likely remains consistent.

Background difference between the ctDCS group and the sham group. I noticed there were some visible differences between the two groups, and I wonder if it is an issue to address. For example, Figure 2I and 2J [pre-stim, KA]. Figure 2K, if your statistics was done on 0-5 Hz, they are different. Figure 3E and F, pre-stim. Figure 4I and K, pre-stim.

Minor

Title

I don't see why enhancing delta oscillation contributes to the anti-epileptic effect. Maybe the tone of the title needs some twists; for example, ...through... => ... with reduced neuronal activity and enhanced delta oscillation.

Methods

Cathodal position requires ML and the depth.

EF measurement: why here? This was not close to the LFP recording site.

Results

3.6, do you have region-specific data for BDNF?

END OF COMMENTS

Chiang, et al., Cathodal weak direct current decreases epileptic excitability through reducing neuronal activity and enhancing delta oscillations

In this study, Chiang et al., applied cathodal direct current stimulation (ctDCS) on an in vivo acute model of seizures, and attempted to uncover the anti-seizing mechanisms of ctDCS, from the aspects of the changes of epileptic spikes, neuronal firing rates, unit-LFP coupling, enhanced inhibition, and c-fos expression. This manuscript provided interesting notions, and the findings may be applicable for clinical usage, but there are still points to clarify:

Major

About the ctDCS. I am sorry if I missed the description, but what were your stimulating patterns? My understanding is: e.g. in Figure 2B, at a current strength of 1 mA, you generated an electrical field from -0.9 to 0.9 mV/mm, and I supposed you switched the polarity (thus you had a positive or a negative field) at 0.5-2 Hz; however, what is the exact length of each polarity? Did you have a short break between each polarity? For how long?

Then, what is the actual pattern of ctDCS used for the rest experiments?

30 minutes of stimulation isn't short; I wonder if your stimuli were good (and stable) for the whole 30 minutes.

Also about the ctDCS itself. Does the ctDCS cause any artifacts? While you made either from 0.5 – 2 Hz ctDCS, I am curious about how the on/off phases of ctDCS affected the LFPs, but I don't see any clues from the recording like Figure 2C.

Different epileptic activities. Chiang et al., described three types of epileptic activities after kainic acid infusion in Results 3.1 and Figure 1; if I did not miss anything important, it seems only 'polyspikes' were used in the following sections. So, I am confused about the need for classification.

Line 302-305, I am not sure I understand the notion.

I think the detected upward events from Figure 2C [Stim] are different from Figure 2C [Post-KA/Pre-stim] or the whole Figure 2C sham group. What was happening? Were these changes in polarities and waveforms common in the ctDCS group? I feel the changes suggest the real ctDCS's effect, and even hint at the involvement of more interneurons (and/or less pyramidal neurons). Maybe you should address this question.

Figure 2D is not very convincing: I would suspect there are still 2 outliers in the sham group at the Stim section; if the two outliers were removed, would they just make the P60-P90 no different?

Figure 2E or G are unclear (the way they are presented), and I cannot grasp anything from the panels. I may not use 'AUC' (AUC is commonly associated with receiver operating characteristic curve) in Figure F and H; what is the reason you wanted to present the area (joules?), but not just the amplitudes? Please specify the purpose.

Unit detection and spike sorting.

It is notoriously difficult to isolate the unit activities from epileptic activities, especially since the authors only had single-channel recordings instead of using tetrodes, which makes it more challenging. One of the important notions in this manuscript is based on the increased/decreased firing rate thus we should be rather careful about false negatives/positives. Using easily changeable measures (i.e. firing frequency) for sorting is risky; would you consider other static measures instead? e.g. descending/ascending slopes, if a trough is following/followed by a positive deflection and then the amplitudes them. Honestly, I felt some firing rates of units are abnormally high (Figure 3B, E, and F; temporally 20-30 Hz might be fine, but continuous 20-30 Hz looks weird) and I could not see a refractory period (~2 ms) from the representative autocorrelation (Figure 3A), thus I worry that couples of units were still not separated.

The raster plots do not reflect the decrease (Figure 3E) and increase (Figure 3F) if the bright bands mean the events.

You may already have a few sorted interneurons, so why not reveal their behaviors? Especially you have mentioned the unit-delta coupling and the potentiated inhibition. On the other hand, I don't feel Figure 3G or 3H is really helpful when the species are different, and I doubt the responses to KA are transferable between species. After spike sorting and checking autocorrelation and cross-correlation, most people would believe your putative pyramidal neurons and interneurons by their mid-width > xxx ms or < xxx ms (Csicsvari et al.).

According to Figure 1G-I or Figure 4D-F, I worry the false negatives were uneven and high from one experiment to another. Were they before or after spike sorting?

Pair-pulse. Figure 5D, this is rather interesting that your P1 changed in the KA-ctDCS rat. As the internal control, P1 likely remains consistent.

Background difference between the ctDCS group and the sham group. I noticed there were some visible differences between the two groups, and I wonder if it is an issue to address. For example, Figure 2I and 2J [pre-stim, KA]. Figure 2K, if your statistics was done on 0-5 Hz, they are different. Figure 3E and F, pre-stim. Figure 4I and K, pre-stim.

Minor

Title

I don't see why enhancing delta oscillation contributes to the anti-epileptic effect. Maybe the tone of the title needs some twists; for example, ...through... => ... with reduced neuronal activity and enhanced delta oscillation.

Methods

Cathodal position requires ML and the depth.

EF measurement: why here? This was not close to the LFP recording site.

Results

3.6, do you have region-specific data for BDNF?

Point-by-Point Response to the Editor's and Reviewers' Comments

Ms. Ref. No.: JP-RP-2024-287969

We sincerely appreciate the editor's and reviewers' valuable comments. In response, we have carefully revised the manuscript and provided a detailed point-by-point reply to each of the comments below. The reviewers' comments are displayed in blue font, while our responses are presented in black. Relevant paragraphs from the revised manuscript are enclosed in double quotation marks, with the changes highlighted in red for clarity.

EDITOR COMMENTS

Reviewing Editor:

Methods Details: Details of analgesia and anaesthesia are needed in methods section 2.6 (optogenetics). Details of any analgesia are missing in section 2.1. The method of euthanasia needs to be specified for all animals used. Please be sure to add these details in order to meet the Journal's ethical standards.

Response: Details of analgesia and anesthesia for each experiment have been provided in Sections 2.1, 2.2, 2.6, and 2.7 of Materials and Methods. The method of euthanasia has been described in Section 2.8.

Section 2.1. “Male Sprague-Dawley rats (BioLASCO), 6 weeks old, were placed at a stereotaxic instrument (RWD) for surgery under anesthesia with 2–2.5% isoflurane inhalation and with pre-surgical analgesia of ketoprofen (2 mg/kg, intraperitoneal).” (Line 139-140)

Section 2.2. “To maintain the stability in unit spike recording during acute seizures, signals were collected under anesthesia with 1–1.5% isoflurane inhalation.” (Line 173-174)

Section 2.6 (optogenetics). “...were unilaterally injected into C57BL/6J male mice, 6–10 weeks old (NCKU animal center) at CA1 (AP: -3, ML: 2.5, DV: -1.6 mm) under anesthesia with 2–2.5% isoflurane inhalation and pre-surgical analgesia with Ketoprofen (3 mg/kg, intraperitoneal).” (Line 248-249)

“The unit recording signals were collected under anesthesia with 1–1.5% isoflurane inhalation at pre-optostim (5min) ...” (Line 256-257)

Section 2.7. “Evoked responses were recorded at the left CA1 (AP: -2.5, ML: -2.5, DV: -3, mm) under anesthesia with 1–1.5% isoflurane inhalation using a tungsten recording microelectrode (127- μ m diameter, A-M system).” (Line 266-267)

Section 2.8. “After completing the experiments, all animals were euthanized using 70% CO₂ administered for at least 5 minutes until the cessation of heartbeat and breathing.” (Line 296-298)

Ethics Concerns: No ethical concerns, but please add the details requested above.

Response: Details regarding analgesia, anesthesia, and euthanasia have been added.

Comments for Authors to ensure the paper complies with the Statistics Policy: Section 2.9 and throughout data analysis - the Journal requires the use of standard deviation rather than SEM - please change this.

Response: Corrected. We have replaced SEM with standard deviation (SD) in Section 2.9 and throughout all data analyses and presentations, including Fig. 2E, G, I, J, K; Fig. 3C, D; Fig. 4J, K; Fig. 5B, C; and Supplementary Fig. 3A, B, C, D.

“Results are reported as median ± interquartile range (IQR) for non-normally distributed data and as mean ± standard deviation (SD) for data that are approximately normally distributed.” (Line 301)

Please report exact p values rather than ranges (e.g. line 326 page 14)

Response: Corrected. All exact p-values have been reported instead of ranges, except in cases where p is less than 0.001, in compliance with the journal's ethical standards.

“post-hoc Tukey's multiple comparisons test, $p=0.006$.” (Line 345)

“In comparison to sham stimulation, the epileptic spikes in ctDCS group decreased during stimulation (unpaired t-test, $p=0.024$) and at 60–90 minutes post-stimulation ($p=0.019$, tDCS $n=11$, sham $n=11$; Fig. 2D).” (Line 351)

“Delta power, normalized to pre-stim level, was significantly higher in the ctDCS group during stimulation ($p=0.001$), 0–30 min ($p=0.024$), 30–60 min ($p=0.003$), and 60–90 min post-stimulation ($p=0.002$) compared to the sham group (Mann-Whitney test, $n=14$; Fig. 2L)” (Line 371-376)

“(Mann-Whitney test, $p=0.001$, $n=3$ mice, Fig. 3H)” (Line 418)

“(Mann–Whitney U test, $p=0.003$, $n=6$)” (Line 477)

“*in e, ≥ 4 to < 5 Hz, ctDCS-stim compared with pre-stim ($p=0.027$), P0–30 ($p=0.014$), P30–60

($p=0.024$), and P60–90 ($p=0.019$), $p<0.05$; $n=14$.” (Line 861-862)

“**in d, ≥ 1 to <2 Hz, sham-stim compared with P0-30, $p=0.001$.” (Line 870)

Comments to the Author: Thank you for submitting your manuscript, which has been reviewed by two expert referees. Although both referees express enthusiasm about this work, they offer a number of comments which they feel will improve the strength of the work.

Response: We greatly appreciate the editor and reviewers for the insightful comments. All feedback has been thoroughly addressed in the revised manuscript, with detailed point-by-point responses provided below.

REFEREE COMMENTS

Referee #1:

The manuscript by Chiang et al., investigates the anti-epileptic mechanism of cathodal weak current transcranial stimulation in an acute rat intra-hippocampal kainate model. For this, under anaesthesia, they apply cathodal transcranial direct current and record the LFP as well as unit activity in the hippocampus while the animal is experiencing a prolonged status epilepticus (the recording/stimulation starts after an initial XX hours). They found that following the stimulation, there is a prolonged (for the next 90min) decrease of neuronal firing associated with an increase of delta power as well as a prolonged paired-pulse depression on top of anti-epileptiform effect which is unseen in the sham stimulation group.

Overall, the manuscript is clear and well-illustrated. The data and mechanisms presented are of importance to the epilepsy field to better understand the advantage and limitations of this transcranial stimulation.

Response: We sincerely appreciate the comments.

Here are my comments that I hope will help consolidate the manuscript:

1. According to the authors and the data, transcranial stimulation would reduce neuronal activity by increasing inhibition since the paired-pulse depression and cfos exp would suggest that there is an increase of inhibition during/after stimulation. It is quite an

attractive explanation, and it would also suggest that the stimulation somehow has a differential effect on different cell types (which is very possible). However, the data presented at least in the cfos experiment are a bit thin since the sample size for GABA neurons co-labeled with cfos is about ~2-4 neurons/um² which is very small. The authors mentioned in the methods spike sorting of interneurons as well. It could greatly strengthen their data if they could analyse the firing pattern of FS interneurons based on the width and ISI. It could help clarify whether some interneurons at least increased their firing during/after the stimulation.

Response: Thank you for your comments. In this study, we investigate the inhibitory effects of cathodal tDCS on seizures, with a particular focus on putative excitatory neurons. We acknowledge the potential differences in response to stimulation between pyramidal cells and interneurons. To address this comment, we also analyzed unit spikes of putative fast-spiking (FS) interneurons, as described below.

“These results indicate that the inhibitory effect of ctDCS on seizures was mediated by reducing activation in CaMKII-labeled excitatory neurons while enhancing activation in GAD-labeled GABAergic inhibitory neurons, showing cell-type-specific responses to ctDCS.” (Line 480-484)

We analyzed unit spikes of putative fast-spiking (FS) interneurons, defined by a narrower spike width (0.3–0.5 ms). In the stratum pyramidale, FS interneuron unit spike signals were not observed consistently through all animals. Only some animals in tDCS and sham groups exhibited a few firing spikes of the putative FS interneurons. These unit spikes were not present across all recording periods during and after stimulation, making it difficult to determine whether interneuron firing rates increased or decreased following stimulation. The sparse interneuron activity observed in unit spike recordings, and the immunostaining expression of c-Fos activation in GABA⁺ neurons, may be related to a compromised inhibitory mechanism in persistent seizures, such as status epilepticus, in this study.

2. The data also suggest that there is a stronger coupling between delta oscillation and neuronal firing in the group receiving the stimulation. The argument being that more delta oscillation more silent firing pause. However, from figure 4i, it seems that there are no preferential oscillation phases of neuronal firing suggesting that there is no more silent period. In addition, are the different units better synchronised between each other?

Response: We appreciate your comments. We have clarified our explanation and revised the discussion to address this point.

Figure 4I (renamed 4J in the revision) illustrates the proportion of unit firing relative to delta

phases, with each spike plotted at its corresponding delta phase and all firing phases aligned shown in a delta cycle. Thus, the phase of unit firing represents its distribution across delta oscillations rather than the exact time intervals between individual spikes. The phase difference is not equivalent to the interval between two spikes. For example, two spikes with a certain phase difference may occur within a single delta cycle or several cycles apart, meaning the inter-spike firing interval is not directly reflected by phase. In phase-locking firing, when neuronal firings concentrate at a particular phase, the non-firing phase could indicate a pause period. However, in a non-phase-preferential pattern, it does not necessarily indicate that the pause period between firings is reduced. In this study, the reduced firing rate, accompanied by stronger coupling to the delta oscillations, suggests that unit firing is tuned to a slower frequency in delta rhythm. This reflects an overall decrease in neuronal excitability and supports the disfacilitatory linkage between unit spikes and delta waves following ctDCS.

“Delta waves are composed of quasi-periodic potential shifts with an upward slow wave consistent with a disfacilitatory event and a pause in spike firing (Buzsaki et al., 2012).

However, this phase-preferential firing pattern would be disrupted in some epileptic animals, varying across hippocampal regions and frequency phases (Shuman et al., 2020). A non-phase-preferential firing of unit spikes to delta oscillation was shown at CA1 of the animals with status epilepticus in the study. This non-phase-preferential firing pattern refers that neurons fire throughout the delta phase but does not necessarily imply a reduction in disfacilitatory periods between firings. Notably, our findings revealed reduced unit spike activity, enhanced delta power (but not in higher frequency bands), and strengthened unit spike-delta wave coupling in ctDCS-treated animals. This strengthened coupling suggests that excitatory neuronal firing is temporally tuned to a slower frequency in delta rhythm, reflecting an overall decrease in neuronal firing excitability and supporting the disfacilitatory linkage between unit spikes and delta waves following ctDCS.” (Line 491-503)

In addition, are the different units better synchronised between each other?

Response: We agree with this and have added an explanation as follows.

“It is possible that different unit spikes synchronize in distinct manners with LFP waves across varying frequency ranges. The unit spike-LFP phase coupling indeed varies between different cell types and is influenced by factors such as cell morphology, perisomatic GABAergic inhibition, and the inputs from intrinsic circuitry (Navas-Olive et al., 2020).” (Line 535-540)

3. All the statistics on single unit are made as the single unit being the experimental unit. It would also be good to see the data per animal either with a colour code or having in the

supp material the same data but within each animal.

Response: Thanks for the comment. We have added **Supplementary Figure 4**, illustrating the unit spike changes for each animal treated with ctDCS or sham stimulation.

“Supplementary Fig. 4. Changes in unit spike numbers for individual animals treated with tDCS or sham stimulation. (A) Unit spike number changes relative to pre-stimulation levels for each KA rat in the ctDCS group, shown across the periods of stim, P0-30, P30-60, and P60-90. (B) Unit spike number changes relative to pre-stimulation levels for each KA rat in the sham-stimulation group, across the periods of stim, P0-30, P30-60, and P60-90. KA, kainic acid; stim, stimulation; P0-30, 0-30 min post-stimulation; P30-60, 30-60 min post-stimulation; P60-90, 60-90 min post-stimulation.” (Line 400 in manuscript; Supplementary Fig. 4. and supplementary figure legend in Line 63-70)

4. Figure 2 E and G are unclear to me. The group stim seems to have more spikes and with a bigger amplitude in all the condition compared to the sham one which is going in the opposite direction of the rest especially with the AUC analysis. In addition, what the dashed lines mean is unclear (min and max and the bold traces are the average?).

Response: We have revised the description in Methods and provided detailed explanation below. The spike amplitude distribution is presented in Figure 2E and G, where the count of spikes for the amplitude is displayed, with the mean represented by solid-colored curves and the **standard deviation (SD)** by dashed lines. (Line 876)

Figure 2E (ctDCS group) and Figure 2G (sham group) illustrates the spike counts under the defined amplitude ranges by area under the curve (AUC). In our study, AUC was calculated as the integral of a function defined by the X-axis (amplitude) and Z-axis (count), where X represents spike amplitude and Z represents the spike count for the corresponding amplitude X.

It is computed using the formula: $AUC = \int_{x_1}^{x_2} f(x)dx$, where $x_1 = 1$, $x_2 = 4$, and $f(x)$ represents

the spike count at the amplitude x . The AUC reflects the distribution of spike amplitudes and their corresponding spike counts. The AUC values for each stage of pre-stimulation, stimulation, and post-stimulation, are displayed on the Y-axis.

The spike count was indeed higher in the ctDCS group compared to the sham group. To address this inter-group difference, we analyzed amplitude changes over time within each group of ctDCS and sham, separately. Additionally, the spike amplitude changes were normalized to the pre-stimulation state for each animal individually, and the during- and post-stimulation changes were compared to the pre-stimulation state within each group, as shown in Figure 2F and 2H.

“The amplitudes of the spikes were calculated by integrating the spike count and amplitude information using the areas under the curve (AUCs) of the spike amplitude distribution, which represents the spike amplitudes and their corresponding spike counts (Fig. 2E and G). The AUC was then normalized to the pre-stimulation stage for each animal individually to compare amplitude changes across the pre-, during, and post-stimulation stages in the ctDCS and sham groups, respectively (Fig. 2F and H).” (Line 194-198)

5. Figure 2 panel K: the power of the low frequencies (~delta power) is quite different between the pre-stim sham and treated group with a lower power for the treated group. It does not seem to be the case in panel I and J why? If it is the case that there is a difference, it should be discussed in term of interpretation of the data.

Response: To address this question, we analyzed the low-frequency power ranging from 0.5 to 5 Hz to compare the pre-stimulation baseline between ctDCS and sham groups. There was no significant difference in the pre-stimulation baseline between groups for the main factor (tDCS vs. sham) in the 0.5–5 Hz frequency range (two-way ANOVA, $F_{(1, 888)}=0.075$, $p=0.785$), as shown in Fig. 2K, although the sham group exhibited relatively higher power than the ctDCS group within the 0.5–1 Hz range. Given this condition, we analyzed the low-frequency power (0.5–5 Hz) changes in tDCS and sham groups separately in Figure 2I and 2J.

We have added the comparison of the pre-stim baseline between ctDCS and sham groups in Results: “The pre-stim baseline did not differ significantly between ctDCS and sham groups in the frequency range of 0.5–5 Hz (two-way ANOVA, $F_{(1, 888)}= 0.075$, $p=0.785$) and 0.5–50 Hz ($p=0.996$; Fig. 2K).” (Line 369-370)

6. Figure 3 panel D: Why the KA-treated group starts below 0 while the sham one starts above? Does it suggest that the decrease observed in the treated group was not already happening before the stim? And the opposite being true for the sham stim? If it is the case

how the authors can distinguish a true effect against a drift?

Response: The number of unit spikes during the stimulation and post-stimulation stages (P0-30, P30-60, P60-90) was subtracted from the pre-stimulation spike count and then divided by the pre-stimulation spike count ($\Delta N/N$) for each individual animal to examine changes in unit spike activity at different stages compared to pre-stimulation. The decreased $\Delta N/N$ observed during the stimulation stage in ctDCS-treated rats indicates a reduction in unit spike activity during stimulation relative to pre-stimulation. Similarly, the decreased $\Delta N/N$ during the post-stimulation stages reflects a sustained reduction in spike activity following stimulation. In contrast, the increased $\Delta N/N$ observed in sham-stimulation animals during both stimulation and post-stimulation stages indicates a progressive increase in neuronal firing over time, reflecting the hyperexcitability characteristic of seizures in the sham stimulation group. The reduction in unit spikes in ctDCS-treated rats compared to the sham-stimulation group supports the effects of ctDCS in reducing neuronal firing in animals with seizures.

“The unit spike number **changes relative to** the pre-stim spikes was significantly reduced by ctDCS (n=14) compared to sham-stimulation (n=14), particularly at P30–60 and P60–90 (two-way ANOVA, $F_{(1, 529)}=108.9$, $p<0.0001$; post-hoc Bonferroni's multiple comparisons test, **$p=0.001$ at stim, $p=0.005$ at P0-30, $p<0.0001$ at P30-60; $p<0.0001$ at P60-90; Fig. 3D and **Supplementary Fig. 4**). The increased unit spikes in sham stimulation reflecting the ongoing epileptic excitation while ctDCS markedly reduced it.” (Line 396-401)**

7. Figure 5 D: in the treated group, post-stimulation you see an increase of P1 while P2 stays stable when compared to pre-stimulation. This would not necessarily suggest an paired-pulse depression but more a potentiation of P1.. Can the authors comment on this and eventually discuss it in the manuscript if true.

Response: We appreciate your comment and have added a paragraph in discussion to address this point.

“Our data indicate that most P1 responses remain stable at P60 and P90 following stimulation, while some P1 responses exhibit an enlarged evoked potential immediately post-ctDCS. Meanwhile, the P1 remains consistent in sham control group. A possible explanation for this P1 potentiation is an increased probability of presynaptic neurotransmitter release and enhanced responsiveness of postsynaptic receptors in certain animals immediately after stimulation. Overall, P2 showed a consistent reduction following ctDCS, and the P2/P1 ratio decreased after ctDCS compared to both the pre-stimulation state and the sham-stimulation group.” (Line 571-578)

8. Figure 6: Are the counting made blind to the condition? It should be indicated in the methods section. In addition, was the entire hippocampus analysed or only sub region? If yes, were the region randomly imaged or not?

Response: Thanks for the questions. We have revised the Methods and provided descriptions regarding the blinded manner in counting and the hippocampal areas sampled for analysis.

“Fluorescence microscopic images for neurons were obtained using an Olympus FluoView FV3000 confocal laser scanning microscope and analyzed using TissueQuest 4.0 **by the operator blinded to experimental conditions.**” (Line 290)

“**The neuronal activations examined by the immunofluorescent staining were sampled from the dorsal hippocampal CA1 and DG below the stimulation cannula (stim-site, AP: -3 mm) and the electrode recording sites (record-site, AP: -5 mm).**” (Line 290-293). The imaging was analyzed for the entire region, defined by an identical area in both groups, and not limited to specific pixels selected from the area, thus reducing selection bias.

“Hippocampal brain-derived neurotrophic factor (BDNF) **collected from dorsal hippocampal area under the stimulation electrode** were analyzed for the concentrations using a conventional ChemiKine BDNF Sandwich ELISA kit (Millipore).” (Line 294-295). No specific subregions were isolated from the sampled tissues for BDNF assay.

9. Supp F1: the frequency definition of each power band would be good to add.

Response: Yes, we do have the definition of each frequency band in the figure legend for Supplementary Figure 1. To make it clear, we also added the definition for each frequency band in Results.

“Delta power (**≥ 0.5 to < 4 Hz**), normalized to pre-stim level, was significantly higher in the ctDCS group during stimulation ($p=0.001$), 0–30 min ($p=0.024$), 30–60 min ($p=0.003$), and 60–90 min post-stimulation ($p=0.002$) compared to the sham group (Mann-Whitney test, $n=14$; Fig. 2L). No significant differences were observed in theta (**≥ 4 to < 8 Hz**), alpha (**≥ 8 to < 12 Hz**), beta (**≥ 12 to < 25 Hz**), and gamma (**≥ 25 to < 50 Hz**) band power between ctDCS and sham groups (Supplementary Fig. 1).” (Line 371-376)

10. Methods: Immuno protocol: what is the blocking solution composition. It should be added to the text.

Response: Corrected. The blocking solution composition is added to Methods.

“Hippocampal coronal sections (40 µm thick) were incubated with primary antibodies ... overnight at 4°C in blocking solution (Dako, #S3022).” (Line 283-284)

11. Is the anaesthesia during the exp the same as for the surgery? 1.5-2% isoflurane? If yes, the authors could add something to make it clear. If not, what was it?

Response: Isoflurane was applied by inhalation at 2–2.5% during surgical anesthesia. It was then tapered to 1–1.5% and maintained throughout the recording period.

“Male Sprague-Dawley rats (BioLASCO), 6 weeks old, were placed at a stereotaxic instrument (RWD) for surgery under anesthesia with 2–2.5% isoflurane inhalation and with pre-surgical analgesia of ketoprofen (2 mg/kg, intraperitoneal).” (Line 139-140)

“To maintain the stability in unit spike recording during acute seizures, signals were collected under anesthesia with 1–1.5% isoflurane inhalation.” (Line 173-174)

12. Line 150: it should read ground instead of "grand".

Response: Corrected. Thank you.

“Two stainless-steel screws (M1, 4 mm length) were placed at ML: 0, AP:-6 for reference and AP: +1.5 mm for ground.” (Line 153)

Referee #2:

Chiang, et al., Cathodal weak direct current decreases epileptic excitability through reducing neuronal activity and enhancing delta oscillations

In this study, Chiang et al., applied cathodal direct current stimulation (ctDCS) on an in vivo acute model of seizures, and attempted to uncover the anti-seizure mechanisms of ctDCS, from the aspects of the changes of epileptic spikes, neuronal firing rates, unit-LFP coupling, enhanced inhibition, and c-fos expression. This manuscript provided interesting notions, and the findings may be applicable for clinical usage, but there are still points to clarify:

Response: We greatly appreciate your comments and have revised the manuscript to address

the points raised.

Major

About the ctDCS. I am sorry if I missed the description, but what were your stimulating patterns? My understanding is: e.g. in Figure 2B, at a current strength of 1 mA, you generated an electrical field from -0.9 to 0.9 mV/mm, and I supposed you switched the polarity (thus you had a positive or a negative field) at 0.5-2 Hz; however, what is the exact length of each polarity? Did you have a short break between each polarity? For how long?

Then, what is the actual pattern of ctDCS used for the rest experiments?

Response: Thank you for the questions. We have revised the manuscript and described details of the stimulation current waveform and polarity for ctDCS, and the alternating current applied for the EF estimation, as follows.

Stimulation pattern of ctDCS: tDCS is a constant direct current without a frequency parameter. The waveform and strength of the ctDCS current applied to animals with KA-induced seizures are illustrated by the red line in the depiction below (Response to Review Figure 1).

“Rats were subjected to **cathodal constant direct current through the epicranial stimulation electrode above dorsal hippocampus with 1 mA (ctDCS) or 0 mA (sham) for 30 min** stimulation with Master-9 stimulator and ISO-Flex stimulus isolator (A.M.P.I.)” (Line 160-161)

Response to Review Figure 1.

Stimulation pattern of alternating current for EF: In Figure 2B, we applied a low-frequency 0.5–2 Hz alternating current to estimate the EF at CA1, induced by the varying current strengths (0.3, 0.5, 0.7, and 1 mA) delivered through the epicranial stimulation electrode. Each stimulation parameter was applied continuously 10 times, with a 5-minute resting interval before switching to the next stimulation parameter. The waveform and polarity changes of the alternating current applied for EF estimation are shown in the depiction below (Response to Review Figure 2).

“Low-frequency sinusoidal stimulations were applied with Stmisola linear isolated stimulator (Biopac Systems Inc.) **through the epicranial stimulation electrode to test the varying transcranial current strength-induced EF at CA1.** The field potentials (mV) were calculated as the difference between the paired electrode tips divided by the distance of 1 mm, resulting in the EF (mV/mm) measured at CA1. (Fig. 2B).” (Line 179-181)

Response to Review Figure 2.

An example of the sinusoidal waveform and polarity of the alternating current used for EF estimation at 0.5 Hz are shown in Figure A, B, C, and D, corresponding to current strengths of 1, 0.7, 0.5, and 0.3 mA, respectively. E1 and E2 represent the electric field potentials recorded from the paired double-wire electrodes with a 1 mm distance at CA1. E2-E1 is the field potentials (mV) calculated as the difference between the paired electrode tips divided by the distance of 1 mm, resulting in the EF (mV/mm) measured at CA1. (Methods as Line 178-183)

30 minutes of stimulation isn't short; I wonder if your stimuli were good (and stable) for the whole 30 minutes.

Response: The stimulation electrical current was verified using a Volt-Ohm-Milliammeter before each experiment to ensure the administrated current strength was consistently

maintained at 1 mA. During stimulation, an indicator light on the stimulator automatically illuminated to confirm that the stimulation output was functioning, and it turned off when the stimulation output was discontinued or terminated.

Also about the ctDCS itself. Does the ctDCS cause any artifacts? While you made either from 0.5 - 2 Hz ctDCS, I am curious about how the on/off phases of ctDCS affected the LFPs, but I don't see any clues from the recording like Figure 2C.

Response: Thank you for the question. ctDCS is a subthreshold weak direct current stimulation. In LFPs, ctDCS induced only transient single artifact spikes at the moments when the stimulation was turned on and off. The morphology of these stimulation-on and stimulation-off induced artifacts was distinctly different from that of epileptic spikes, and these artifacts were excluded from the analysis. The LFPs presented in Figure 2C illustrate the LFP with epileptic spikes during 1mA-ctDCS. The transient artifacts induced by ctDCS stimulation-on and stimulation-off are shown alongside the LFPs recorded during ctDCS in the following depiction (Response to Review Figure 3).

Response to Review Figure 3.

Otherwise, no stimulation artifacts were observed during the entire stimulation period. “The signals were clear without stimulation artifacts from the weak direct current stimulation, for which no additional template subtraction for artifacts was implemented.” (Line 189-191)

Different epileptic activities. Chiang et al., described three types of epileptic activities after kainic acid infusion in Results 3.1 and Figure 1; if I did not miss anything important, it seems only 'polyspikes' were used in the following sections. So, I am confused about the

need for classification.

Response: Thanks for the question. We have provided revision and description as follows.

“Epileptic activities appear in distinct patterns during seizure evolution following KA induction, prior to reaching status epilepticus. Once status epilepticus is achieved, the LFPs predominantly exhibit sustained high-frequency polyspikes.” (Line 312-315)

In our study, we analyzed signals primarily composed of polyspikes during status epilepticus to investigate the effects of ctDCS on these sustained epileptic activities. However, we think it is important to illustrate the epileptiform discharges in LFPs and their relationship with unit spike activities during seizure evolution. The relationship observed between unit spikes and epileptiform discharges further strengthens the rationale for the subsequent study, suggesting that modulating unit spike activity can influence the epileptic LFP.

Line 302-305, I am not sure I understand the notion.

Response: Thank you for pointing out this notion. We’ve revised it. The result in Figure 1 indicates that the unit activity is coupled to the epileptiform discharges observed in LFPs. This observation provides the rationale for applying ctDCS in animals with seizure to investigate its effects on unit spike, LFPs and their relationship in this study.

“This finding raises the hypothesis that epileptic LFPs can be modulated by altering unit spike activity. To investigate this, we examine unit spikes, LFP oscillations, and their coupling in response to neuromodulation by ctDCS.” (Line 327-330)

I think the detected upward events from Figure 2C [Stim] are different from Figure 2C [Post-KA/Pre-stim] or the whole Figure 2C sham group. What was happening? Were these changes in polarities and waveforms common in the ctDCS group? I feel the changes suggest the real ctDCS's effect, and even hint at the involvement of more interneurons (and/or less pyramidal neurons). Maybe you should address this question.

Response: We appreciate your insightful comment. We have addressed this notion in Discussion.

“Interesting, a polarity change in epileptic spikes within LFPs was observed during and following ctDCS compared to pre-stim (Fig. 2C). Based on current flow measurement (Buzsaki et al., 2012), we speculate that this polarity shift may indicate the involvement of inhibitory events, where a change in current flow from inward to outward in the neuronal population suggests a hyperpolarization shift from depolarization. However, this polarity change in LFPs

during stimulation was not observed in all animals. Most animals exhibited the same polarity of epileptic spikes in LFPs. Investigating the physiological mechanisms underlying these polarity changes in epileptic field potentials during neuromodulation would be notable for future research.” (Line 508-517)

Figure 2D is not very convincing: I would suspect there are still 2 outliers in the sham group at the Stim section; if the two outliers were removed, would they just make the P60-P90 no different?

Response: Thank you for the question. We clarify that there were no statistical outliers in Figure 2D. An outlier was defined as a value exceeding the mean \pm two standard deviations (SD). The values observed in the upper range of the Stim section (1.54 and 1.59; group mean: 0.145, SD: 0.850, mean \pm 2SD: -1.555 to 1.845) and P60-90 (0.19 and 0.10; group mean: -0.524, SD: 0.470, mean \pm 2SD: -1.465 to 0.416) in the sham-stim group were within this range and, therefore, not statistical outliers.

Figure 2E or G are unclear (the way they are presented), and I cannot grasp anything from the panels. I may not use 'AUC' (AUC is commonly associated with receiver operating characteristic curve) in Figure F and H; what is the reason you wanted to present the area (joules?), but not just the amplitudes? Please specify the purpose.

Response: Thank you for the question. We have revised it in Method and provided a detailed explanation as follows.

In our study, AUC was calculated as the integral of a function defined by the X-axis (amplitude) and Z-axis (count), where X represents spike amplitude and Z represents the spike count for the corresponding amplitude X. The AUC in Figure 2E (ctDCS group) and Figure 2G (sham group) illustrates the spike counts under the defined amplitude ranges. It is computed using the

formula: $AUC = \int_{x_1}^{x_2} f(x)dx$, where $x_1 = 1$, $x_2 = 4$, and $f(x)$ represents the spike count at the amplitude x . The AUC reflects the distribution of spike amplitudes and their corresponding spike counts. The AUC values for each stage of pre-stimulation, stimulation, and post-stimulation, are displayed on the Y-axis.

“The amplitudes of the spikes were calculated by integrating the spike count and amplitude information using the areas under the curve (AUCs) of the spike amplitude distribution, which represents the spike amplitudes and their corresponding spike counts (Fig. 2E and G). The AUC was then normalized to the pre-stimulation stage for each animal individually to compare

amplitude changes across the pre-, during, and post-stimulation stages in the ctDCS and sham groups, respectively (Fig. 2F and H).” (Line 194-198)

Unit detection and spike sorting.

It is notoriously difficult to isolate the unit activities from epileptic activities, especially since the authors only had single-channel recordings instead of using tetrodes, which makes it more challenging. One of the important notions in this manuscript is based on the increased/decreased firing rate thus we should be rather careful about false negatives/positives. Using easily changeable measures (i.e. firing frequency) for sorting is risky; would you consider other static measures instead? e.g. descending/ascending slopes, if a trough is following/followed by a positive deflection and then the amplitudes them. Honestly, I felt some firing rates of units are abnormally high (Figure 3B, E, and F; temporally 20-30 Hz might be fine, but continuous 20-30 Hz looks weird) and I could not see a refractory period (~2 ms) from the representative autocorrelation (Figure 3A), thus I worry that couples of units were still not separated.

It is notoriously difficult to isolate the unit activities from epileptic activities, especially since the authors only had single-channel recordings instead of using tetrodes, which makes it more challenging.

Response: Thank you for the comments. We isolated unit spikes from epileptic activities in the LFPs primarily based on the distinct waveform differences between these two signals. LFP epileptic spikes were identified using the following criteria: “Peaks with amplitudes greater than 10 times the baseline average and an absolute amplitude ≥ 1 mV and width ≤ 140 ms were labeled as epileptic spikes in LFPs using epileptic spike detection algorithm”. In contrast, unit spikes have much narrower spike widths (0.5–1 ms, selected as putative excitatory neurons) and smaller amplitudes (approximately 0.054 mV) compared to LFP spikes.

We agree that tetrodes allow for more precise spike sorting in the hippocampus. However, due to the limited skull space available for implanting both the stimulation setup and the recording headstage, we opted for single-channel recording in this study. We acknowledge this limitation and have addressed it in the discussion: “We used single-channel recording instead of tetrodes due to the limited skull space available for implanting the stimulation setup and recording electrode headstage.” (Line 607-609)

One of the important notions in this manuscript is based on the increased/decreased firing rate thus we should be rather careful about false negatives/positives. Using easily changeable measures (i.e. firing frequency) for sorting is risky; would you consider other static measures instead? e.g. descending/ascending slopes, if a trough is following/followed by a positive deflection and then the amplitudes them.

Response: We identified and sorted the unit spikes as the follows: “The unit spikes were sorted using the UltraMegaSort 2000 algorithm based on waveform similarity through k-means clustering and firing features analysis, incorporating autocorrelation and cross-correlation (Hill et al., 2011). The output was then imported into MATLAB, where clusters were manually edited using custom spike sorting algorithms based on waveform similarity and spike width.” (Line 216-219)

We agree with your point about the variability of unit spike firing frequency, particularly in seizures. We did not use firing frequency as a sorting criterion. Instead, the firing frequency shown in Fig. 3 illustrates its distribution in animals with seizures. In addition to spike width and amplitude, we also analyzed the descending and ascending slopes of the identified unit spikes as a static measure, as you mentioned: “mean value, peak-to-peak spike-width 0.75 ms, amplitude 54 μ V, left descending slope -0.15 mV/ms, right ascending slope 0.13 mV/ms” (Fig. 3A, Line 385-386).

Honestly, I felt some firing rates of units are abnormally high (Figure 3B, E, and F; temporally 20-30 Hz might be fine, but continuous 20-30 Hz looks weird) and I could not see a refractory period (~2 ms) from the representative autocorrelation (Figure 3A), thus I worry that couples of units were still not separated.

Response: We have provided detailed descriptions of Figure 3B, E and F, and have modified Figure 3A to enhance the clarity of the autocorrelogram.

The firing rates in Fig. 3B, E, and F are presented as the mean value of the collected unit spikes from each 2-minute segment within the selected periods. “Signals for unit spike sorting were sampled from 3–5, 8–10, 13–15, 18–20, 23–25, and 28–30 min over each period of pre-stim, stim, and post-stim in ctDCS- and sham-treated animals.” (Line 211-213). Thus, the increased firing frequency up to or above 20 Hz was observed temporarily within the sampled 2-minute periods but was not persistent throughout the entire recording period. Indeed, increased unit firings with varying degrees of variability were observed during seizures. “The firing frequency in some neurons accelerates 2–3 times during ictal period than the baseline and decreases by half in post-ictal period (Alvarado-Rojas et al., 2013; Neumann et al., 2017).” (Line 524-525).

Due to the compressed time axis (-100 to +100 ms) and the red color bar indicating the

refractory period, the firing activity during this period was not clearly illustrated in the autocorrelogram shown in Fig. 3A. To address this, we have removed the red color bar (Fig. 3A, left panel in the depiction below) and zoomed in on the autocorrelogram to a time scale of -10 to +10 ms (right panel in the depiction below, Response to Review Figure 4). This adjustment demonstrates no unit spike firing within ± 1 ms and only very rare firing occurrences within 1–2 ms.

Response to Review Figure 4.

You may already have a few sorted interneurons, so why not reveal their behaviors? Especially you have mentioned the unit-delta coupling and the potentiated inhibition. On the other hand, I don't feel Figure 3G or 3H is really helpful when the species are different, and I doubt the responses to KA are transferable between species. After spike sorting and checking autocorrelation and cross-correlation, most people would believe your putative pyramidal neurons and interneurons by their mid-width > xxx ms or < xxx ms (Csicsvari et al.).

Response: Thank you for your comments. To address these points, we have provided a description of spike sorting for interneurons and included a statement in the limitation section regarding the transferability of KA between species.

In this study, we investigate the inhibitory effects of cathodal tDCS, with a particular focus on putative excitatory neurons. To address this interneuron firing changes, we analyzed unit spikes of putative fast-spiking (FS) interneurons, defined by a narrower spike width (0.3–0.5 ms). In the stratum pyramidale, FS interneuron unit spikes were not observed consistently through all animals. Only some animals in tDCS and sham groups exhibited a few firing spikes of the putative FS interneurons. These unit spikes were not present across all recording periods during and after stimulation, making it difficult to determine whether interneuron firing rates and the phase coupling increased or decreased following stimulation. The sparse interneuron activity observed in unit spike recordings, and the immunostaining expression of c-Fos activation in GABA⁺ neurons, may be related to a compromised inhibitory mechanism in persistent seizures,

such as status epilepticus, in this study.

“Taking advantage of optogenetic manipulation, transgenic mice with KA-induced seizures were used to verify the cell type of the sorted unit spikes. However, the transferability of KA-induced seizures and unit spikes between mice and rats should still be considered.” (Line 609-612)

According to Figure 1G-I or Figure 4D-F, I worry the false negatives were uneven and high from one experiment to another. Were they before or after spike sorting?

Response: The raw traces sampled for unit spikes are presented at the bottom (before sorting) of Figure 1G–I and Figure 4D–F, while the sorted spikes are shown in the upper panels (after sorting). Some spiking signals visible in the raw trace were not identified as the unit spikes because they did not meet the selection criteria for spike width or waveform pattern.

Pair-pulse. Figure 5D, this is rather interesting that your P1 changed in the KA-ctDCS rat. As the internal control, P1 likely remains consistent.

Response: We appreciate this comment. We have added a paragraph in Discussion to address this point. “Our data indicate that most P1 responses remain stable at P60 and P90 following stimulation, while some P1 responses exhibit an enlarged evoked potential immediately post-ctDCS. Meanwhile, the P1 remains consistent in sham control group. A possible explanation for this P1 potentiation is an increased probability of presynaptic neurotransmitter release and enhanced responsiveness of postsynaptic receptors in certain animals immediately after stimulation. Overall, P2 showed a consistent reduction following ctDCS, and the P2/P1 ratio decreased after ctDCS compared to both the pre-stimulation state and the sham-stimulation group.” (Line 571-578)

Background difference between the ctDCS group and the sham group. I noticed there were some visible differences between the two groups, and I wonder if it is an issue to address. For example, Figure 2I and 2J [pre-stim, KA]. Figure 2K, if your statistics was done on 0-5 Hz, they are different. Figure 3E and F, pre-stim. Figure 4I and K, pre-stim.

Response: Thank you for the comments. We have included the pre-stimulation baseline comparison of tDCS and sham group in Results and descriptions as follows. The pre-stimulation backgrounds between ctDCS and sham were not significantly different in

Figure 2I, J, K (0.5–5 Hz), Figure 3E, F, and Figure 4I (renamed 4J in the revision) and K.

Figure 2I, 2J, 3E, 3F, 4J, and 4K represent within-group analyses assessing the stimulation effects of tDCS or sham over time; therefore, the pre-stimulation comparison between tDCS and sham groups does not influence these within-group analyses. For inter-group comparisons of tDCS and sham, the spike number changes were relative to the pre-stimulation state for each animal individually. As a result, baseline variations between individual animals or groups do not affect the inter-group comparisons. Statistical analysis and description are as follows to address each point raised.

In **Figure 2**, we analyzed the low-frequency power (0.5–5 Hz) to compare the pre-stimulation baseline between ctDCS and sham groups. “The pre-stim baseline did not differ significantly between ctDCS and sham groups in the frequency range of 0.5–5 Hz (two-way ANOVA, $F_{(1, 888)}=0.075$, $p=0.785$) and 0.5–50 Hz ($p=0.996$; Fig. 2K).” (Line 369-370). Therefore, we analyzed the low-frequency power (0.5–5 Hz) changes within each group (ctDCS and sham) separately in Figure 2I and J. The baseline comparison between tDCS and sham groups does not affect the presented analyses.

For **Figure 3E and F**, the unit spike numbers at the pre-stimulation stage showed no significant difference between the ctDCS and sham groups (Mann Whitney test, $p=0.402$). For **Figure 4J and K**, the proportions of the unit spike to delta phase at pre-stim stage also showed no significant difference between tDCS and sham groups (two-way ANOVA, $F_{(1, 2363)}=2.282$, $P=0.131$). Furthermore, we compared these values within each group across pre-stim, stim, and post-stim stages. The baseline comparison between tDCS and sham groups does not affect the presented results.

Minor

Title: I don't see why enhancing delta oscillation contributes to the anti-epileptic effect. Maybe the tone of the title needs some twists; for example, ...through... => ... with reduced neuronal activity and enhanced delta oscillation.

Response: We appreciate your comment and have amended the title to “Cathodal weak direct current decreases epileptic excitability with reduced neuronal activity and enhanced delta oscillations.” (Line 1-2)

Methods: Cathodal position requires ML and the depth.

Response: To simulate non-invasive brain stimulation, cathodal stimulation was administered through an epicranial plastic cannula (1 cm in height, 1.59 mm inner diameter) affixed to the skull at an anteroposterior (AP) coordinate of -3 mm. The cannula was filled with 0.9% normal saline, serving as a conductive medium for the cathodal stimulation electrode (a 0.6-mm diameter gold-plated copper alloy pin electrode, Nan E. Electronic Co.) (Line 140-144, and Figure 2C). As the cathodal electrode was secured on the skull without penetrating the brain, no corresponding mediolateral (ML) or depth coordinates were applicable for this epicranial electrode.

EF measurement: why here? This was not close to the LFP recording site.

Response: We selected this EF measurement location to capture the induced EF at the nearest possible site to the stimulation electrode. The electrode position for EF measurement was located in the dorsal hippocampus CA1 (AP: -3 mm, ML: 1.5 mm, DV: -3 mm and -2 mm). This measurement site is in close proximity to the stimulation electrode cannula, which was fixed to the skull with its center at AP: -3 mm. We believe that the induced EF encompasses the CA1 LFP recording area (AP: -5 mm, ML: 3.5 mm, DV: -3 mm), with a spatial gradient relative to the EF measurement site.

Results: 3.6, do you have region-specific data for BDNF?

Response: We collected dorsal hippocampal tissues from beneath the stimulation electrode for BDNF assay and comparison between ctDCS and sham treatments; however, we did not isolate specific subregions. “Hippocampal brain-derived neurotrophic factor (BDNF) **collected from dorsal hippocampal area under the stimulation electrode** were analyzed for the concentrations using a conventional ChemiKine BDNF Sandwich ELISA kit (Millipore)” (Line 294-295)

END OF COMMENTS

Dear Dr Wu,

Re: JP-RP-2025-287969R1 "**Cathodal weak direct current decreases epileptic excitability with reduced neuronal activity and enhanced delta oscillations**" by Chia-Chu Chiang, Miao-Er Chien, Yu-Chieh Huang, Jyun-Ting Lin, Sheng-Fu Liang, Kuei-Sen Hsu, Dominique M Durand, and Yi-Jen Wu

Thank you for submitting your manuscript to The Journal of Physiology. It has been assessed by a Reviewing Editor and by 2 expert referees and we are pleased to tell you that it is acceptable for publication following satisfactory revision.

REVISION CHECKLIST:

- 'Potential Cover Art' for consideration as the issue's cover image
- Appropriate Supporting Information (Video, audio or data set: see https://jpp.msubmit.net/cgi-bin/main.plex?form_type=display_requirements#supp).

We look forward to receiving your revised submission.

Yours sincerely,

Katalin Toth
Senior Editor
The Journal of Physiology

EDITOR COMMENTS

Reviewing Editor:

Methods Details:

I could not find information on animals access to food and water - please can you add this information to the methods section?

Comments to the Author:

Thank you for submitting a revised manuscript. Both referees agree that the revisions made have further enhanced the manuscript. Referee 2 has some very minor remaining concerns to address. I also could not find information on animals' access to food and water within the methods section - please add this as it is required by J Physiol.

REFEREE COMMENTS

Referee #1:

Many thanks for answering my comments. I have nothing to add.

Referee #2:

I appreciate the authors' responses, and my previous concerns were addressed.

The following are minor:

1) The authors show cathodal direct current reduces epileptic events using the integral of amplitudes and counts from different groups (Figure 2F and H). It is a good practice to present the data (Figure 2E and G), however, the 2E and G are not helpful, to be honest. As an audience, I want to "look very close" at the redistribution due to the cathodal direct current. Did the counts and amplitudes reduce at the same time? or just the counts or the amplitude? that I cannot tell from these two panels.

I suggest authors use two-dimensional plots, and they would be similarly complicated as Figure 2I and 2J but satisfy our curiosity. Besides, what are those small and lighter bars, please describe them.

2) Please check all your n numbers: I would rather have "11 rats" and/or 34567 cells but not n=11 or n=34567. I lost the track, but somewhere (maybe multiple sites) the rat numbers were not provided.

END OF COMMENTS

Point-by-Point Response to the Editor's and Reviewers' Comments

Ms. Ref. No.: JP-RP-2024-287969

We sincerely appreciate the editor's and reviewers' valuable comments. We have carefully revised the manuscript (JP-RP-2024-287969R2) and provided a detailed point-by-point response to each of the comments. The reviewers' comments are displayed in blue font, while our responses are presented in black. Relevant paragraphs from the revised manuscript are enclosed in double quotation marks, with the changes highlighted in red.

EDITOR COMMENTS

Reviewing Editor:

Methods Details:

I could not find information on animals access to food and water - please can you add this information to the methods section?

Response: We appreciate your feedback. To address this, we have included the following description in the Materials and Methods section:

“The experimental animals were housed with ad libitum access to food and water.” (Line 163-164)

Comments to the Author:

Thank you for submitting a revised manuscript. Both referees agree that the revisions made have further enhanced the manuscript. Referee 2 has some very minor remaining concerns to address. I also could not find information on animals' access to food and water within the methods section - please add this as it is required by J Physiol.

Response: We sincerely appreciate all the comments from editor and reviewers. The comments from editor and Referee 2 have been fully addressed in the revised manuscript, with detailed point-by-point responses.

REFeree COMMENTS

Referee #1:

Many thanks for answering my comments. I have nothing to add.

Response: We greatly appreciate your comments!

Referee #2:

I appreciate the authors' responses, and my previous concerns were addressed.

The following are minor:

1) The authors show cathodal direct current reduces epileptic events using the integral of amplitudes and counts from different groups (Figure 2F and H). It is a good practice to present the data (Figure 2E and G), however, the 2E and G are not helpful, to be honest. As an audience, I want to "look very close" at the redistribution due to the cathodal direct current. Did the counts and amplitudes reduce at the same time? or just the counts or the amplitude? that I cannot tell from these two panels.

Response: Thank you for the question. Regarding the effect of ctDCS on epileptic spike amplitude distribution, the spike counts within the defined amplitude ranges decreased over time (Fig 2E, two-way ANOVA, stim-period: $F(4, 29196) = 10.79, P < 0.0001$; post-hoc Dunn's multiple comparison: significant reduction at P0–30, P30–60, and P60–90 vs. pre-stim, $P < 0.0001$). Additionally, spike amplitudes (Fig. 2F) also showed a significant reduction across time in the ctDCS group but not in the sham group, as described below.

“a significant decrease in LFP spike amplitude after ctDCS lasting up to 90 minutes post-stimulation (Kruskal-Wallis test, stim-period $p < 0.0001$; post-hoc Dunn's multiple comparisons test, when compared to pre-stim, $p = 0.004$ at P0–30, $p = 0.002$ at P30–60, and $p < 0.0001$ at P60–90; compared to stim, $p = 0.005$ at P60–90; tDCS: 11 rats; Fig. 2E and 2F)” (Line 355-359)

I suggest authors use two-dimensional plots, and they would be similarly complicated as Figure 2I and 2J but satisfy our curiosity. Besides, what are those small and lighter bars, please describe them.

Response: Thanks for the suggestion. To address it, the spike counts of the defined amplitude ranges were presented in a two-dimensional visualization format (similar to Figure 2I and 2J, shown below in Response to Review Figure 1). However, due to significant overlap among recording periods, we opted for a three-dimensional representation to more clearly illustrate the amplitude distribution over time.

Response to Review Figure 1. Epileptic spike counts of the defined amplitude ranges in ctDCS- (left panel) and sham-treated (right panel) groups.

Besides, what are those small and lighter bars, please describe them.

Response: We appreciate your attention to detail. The small, lighter bars in Fig. 2E and 2G represent standard deviation (SD). While this was previously noted in the figure legend, we have now revised the description for improved clarity.

Original Legend: “Color curves with bands, mean with SD in Fig. 2E, G...”

Revised Legend: “ **Solid color curves with lighter dashed lines, mean with SD in Fig. 2E and G.**” (Line 876-877)

2) Please check all your n numbers: I would rather have "11 rats" and/or 34567 cells but not n=11 or n=34567. I lost the track, but somewhere (maybe multiple sites) the rat numbers were not provided.

Response: We appreciate this suggestion and have carefully checked all sample size notations for both animal numbers and unit spike numbers. Throughout the text, we now present sample sizes as “tDCS: 14 rats” and “unit spike numbers: 53,454” instead of “n = 14” or “n = 53,454”, respectively. However, to maintain conciseness in figure legends, we continue using 'n' with the explicit notation 'n, animal number' for clarity.”

Changes include examples such as “**tDCS: 14 rats, sham: 14 rats**” (Line 376) and “**unit spike**

numbers: pre-stim: 53,454” (Line 433), and so on, among others.

END OF COMMENTS EDITOR COMMENTS

Dear Dr Wu,

Re: JP-RP-2025-287969R2 "**Cathodal weak direct current decreases epileptic excitability with reduced neuronal activity and enhanced delta oscillations**" by Chia-Chu Chiang, Miao-Er Chien, Yu-Chieh Huang, Jyun-Ting Lin, Sheng-Fu Liang, Kuei-Sen Hsu, Dominique M Durand, and Yi-Jen Wu

We are pleased to tell you that your paper has been accepted for publication in The Journal of Physiology.

Yours sincerely,

Katalin Toth
Senior Editor
The Journal of Physiology

If you would like to receive our 'Research Roundup', a monthly newsletter highlighting the cutting-edge research published in The Physiological Society's family of journals (The Journal of Physiology, Experimental Physiology, Physiological Reports, The Journal of Nutritional Physiology and The Journal of Precision Medicine: Health and Disease), please click this link, fill in your name and email address and select 'Research Roundup':
<https://www.physoc.org/journals-and-media/membernews>

- You can help your research get the attention it deserves! Check out Wiley's free Promotion Guide for best-practice recommendations for promoting your work at: www.wileyauthors.com/eeo/guide. You can learn more about Wiley Editing Services which offers professional video, design, and writing services to create shareable video abstracts, infographics, conference posters, lay summaries, and research news stories for your research at: www.wileyauthors.com/eeo/promotion.

EDITOR COMMENTS

Reviewing Editor:

Comments to the Author:

Thank you for addressing the final comments.